materials science/nanotechnology/environmental engineering

cellulose nanocrystal, graphene oxide, adsorption, levofloxacin hydrochloride, antibiotic

**Author for correspondence:**
Junfeng Li
e-mail: ljfshz@126.com

This article has been edited by the Royal Society of Chemistry, including the commissioning, peer review process and editorial aspects up to the point of acceptance.

# Cellulose nanocrystals/graphene oxide composite for the adsorption and removal of levofloxacin hydrochloride antibiotic from aqueous solution

Junhong Tao[1], Jie Yang[1], Chengxiao Ma[1], Junfeng Li[1], Keqing Du[1], Zhen Wei[1], Cuizhong Chen[1], Zhaoyang Wang[2,3], Chun Zhao[1,3,4] and Xiaoya Deng[5]

[1]College of Water Conservancy and Architecture Engineering, Shihezi University, Shihezi 832000, Xinjiang, People's Republic of China
[2]College of Earth and Environmental Science, Lanzhou University, Lanzhou 730000, Gansu, People's Republic of China
[3]School of Urban Construction and Environmental Engineering, Chongqing University, 400001, Chongqing, People's Republic of China
[4]Key laboratory of the Three Gorges Reservoir Region's Eco-Environment, Ministry of Education, Chongqing University, 400045, Chongqing, People's Republic of China
[5]State Key Laboratory of Simulation and Regulation of Water Cycle in River Basin, China Institute of Water Resources and Hydropower Research, Beijing, 100038, People's Republic of China

JL, 0000-0002-4447-0800

Residual antibiotics in water are often persistent organic pollutants. The purpose of this study was to prepare a cellulose nanocrystals/graphene oxide composite (CNCs-GO) with a three-dimensional structure for the removal of the antibiotic levofloxacin hydrochloride (Levo-HCl) in water by adsorption. The scanning electron microscope, Fourier transform infrared (FT-IR), energy-dispersive spectroscopy, X-ray photoelectron spectroscopy and other characterization methods were used to study the physical structure and chemical properties of the CNCs-GO. The three-dimensional structure of the composite material rendered a high surface area and electrostatic attraction, resulting in increased adsorption capacity of the CNCs-GO for Levo-HCl. Based on the Box–Behnken design, the effects of different factors on the removal of Levo-HCl by the CNCs-GO were explored. The composite material exhibited good antibiotic adsorption

capacity, with a removal percentage exceeding 80.1% at an optimal pH of 4, the adsorbent dosage of 1.0 g l$^{-1}$, initial pollutant concentration of 10.0 mg l$^{-1}$ and contact time of 4 h. The adsorption isotherm was well fitted by the Sips model, and kinetics studies demonstrated that the adsorption process conformed to a quasi-second-order kinetics model. Consequently, the as-synthesized CNCs-GO demonstrates good potential for the effective removal of antibiotics such as levofloxacin hydrochloride from aqueous media.

## 1. Introduction

Water-soluble antibiotics constitute a serious type of organic environmental pollutants as the early lack of guidance and regulatory measures related to antibiotics has led to their overuse and abuse [1]. Once released into the environment, these substances spread through aquatic systems and the food chain where they persist and become enriched in the air, water and soil [2,3]. The heavy use of antibiotics has exerted a huge impact on ecosystems and human health, and the problem of antibiotic removal urgently requires attention [4,5].

In recent years, a variety of methods have been explored to mitigate environmental antibiotic contamination, such as adsorption, catalytic degradation, biodegradation, photocatalytic degradation and advanced oxidation [1,6–8]. Among these methods, the widely used adsorption technique offers the advantages of easy operation, flexibility, low energy consumption, high removal rates, low secondary pollution and low adsorbent regeneration cost [9]. Many absorbent materials have been explored for the removal of antibiotics from water, including activated carbon [10], anaerobic granular sludge [11], kaolin [12], graphene oxide (GO) [13], multi-walled carbon nanotubes [14], fly ash [15], montmorillonite [16] and bamboo charcoal [17]. However, the inadequate adsorption efficiencies, poor adsorption capacities, unsatisfactory recyclability, secondary pollution problems and high costs of these adsorbent materials have greatly hampered their practical application. Therefore, the development of new, high-efficiency, low-cost adsorbents for the removal of antibiotics from water is a pressing need.

One material that meets some of the requirements for a good adsorbent is cellulose, an abundant natural resource. Its structural formula reveals a plethora of exposed hydroxyl and reduced and non-reduced end groups at which chemical reactions can occur. As a modified derivative of cellulose, nanocellulose [18] is a new biomass nanomaterial with attractive characteristics such as a large specific surface area, high mechanical strength, good thermal stability and low thermal expansion coefficient because of its highly reduced particle size [19]. Due to the large number of hydroxyl groups on the surface of nanocellulose (table 1), it is easily modified by reactions such as carboxylation, oxidation, esterification, etherification, silanization, acetylation and graft polymerization. These modifications facilitate its addition and dispersal into different polymers [20–23]. Hence, nanocellulose nanocomposites have been used as sorbents for heavy metals and organic pollutants removal from aqueous solution [24]. It is known that graphene has many excellent characteristics such as high strength, high intrinsic mobility, high specific surface area, high light transmittance and high thermal conductivity [25]. Among graphene materials, graphene oxide (table 1) is a commonly used type. It is structurally similar to graphene. That is, the bottom surface of graphene contains epoxy groups and hydroxyl groups, and the edges of graphene have carboxyl groups and carbonyl groups [13,26]. These hydrophilic groups impart surface activity and wettability, and their electrostatic repulsion characteristics facilitate the stable dispersion of GO in water or alkaline solution [27–29]. This stability has an important influence on the mechanical strength and electrical properties of the GO, and also allows various functional groups to be grafted onto modified GO sheets, expanding their prospective range of applications. Recently, GO has attracted increasing attention as a new adsorbent owing to its distinguished properties of high surface area as well as easy to functionalize ability [30]. Response surface methodology (RSM) is a mathematical statistical method for solving multi-variable problems by using reasonable experimental design methods and experimental data, fitting the functional relationship between various factors and response values with multiple quadratic regression equations and obtaining the optimal process parameters through regression analysis [31,32]. RSM overcomes the disadvantage of orthogonal design, which can only deal with discrete level values. It reduces experiment time, offers high precision and exhibits good predictive performance [33]. It is widely used in the experimental design of food, chemical, biological and other studies, but is rarely used in water pollution control and theoretical research.

**Table 1.** The molecular structures of main medicine.

| graphene oxide | nanocellulose | levofloxacin hydrochloride |

The purpose of the present work is the synthesis of novel adsorbent, cellulose nanocrystals/graphene oxide nanocomposite (CNCs-GO), for the removal of antibiotic levofloxacin hydrochloride (Levo-HCl, table 1), a pharmaceutical contaminant in waste water treatment. Adsorbent was characterized using Fourier transform infrared (FT-IR), X-ray diffraction (XRD), scanning electron microscope (SEM)–energy-dispersive spectroscopy (EDS), X-ray photoelectron spectroscopy (XPS) and Brunauer–Emmett–Teller (BET) analysis. The response surface method (RSM) was used to optimize Levo-HCl adsorption conditions. Finally, we investigated the adsorption performance and mechanism for Levo-HCl removal process by CNCs-GO.

# 2. Material and methods

## 2.1. Materials

Levofloxacin hydrochloride 98%, cellulose nanocrystals and graphite powder 99.95% (for the preparation of GO) were purchased from Shanghai Macklin Biochemical Co., Ltd. Deionized water was used for all experiments. Otherwise specified chemicals were of reagent grade and used without further purification. Deionized water was used throughout the experiments.

## 2.2. Preparation of nanocrystalline cellulose/graphene oxide composite

GO was prepared from graphite powder according to the Hummers method [34]. To prepare the composite, the GO suspension ($5 \text{ g l}^{-1}$, 120 ml) was accurately measured into a 500 ml round-bottomed flask, followed by the CNCs suspension ($14.434 \text{ g l}^{-1}$, 100 ml). The brown mixture was continuously sonicated for 1 h, and then stirred at room temperature for 2 h to obtain the brown CNCs-GO suspension. This suspension was freeze-dried (Christ Alpha 1–2 LDplus, Germany) for 48 h to produce the CNCs-GO composite.

## 2.3. Characterization

The FT-IR spectra of CNCs, GO and CNCs-GO samples were recorded using a Nicolet 6700 FT-IR instrument (USA). The spectra were recorded over the wave number range of $400–4000 \text{ cm}^{-1}$ with 10 scans at a resolution of $4 \text{ cm}^{-1}$. CNCs, GO and CNCs-GO were performed to analyse the surface morphology using Quanta 650FEG Emission SEM (USA) and 10 kV accelerating voltage. The specific surface area and the pore volume of CNCs, GO and CNCs-GO were measured using ASAP 2020 Surface and porosity analyser (USA) under $N_2$ analysis adsorptive at 77 K. Powered XRD patterns were obtained (Bruker D8 X-ray powder diffractrometer, Germany) using Cu-K$\alpha$ radiation ($\lambda = 0.15418$ nm). XPS were performed with an Al K$\alpha$ mono and vacuum degree of the analysis room 10–8 mbar (ThermoScientific Escalab250Xi, USA). The contact angle tester (LSA100, LAUDA Scientific, Germany) was used to record the dynamic contact angles of CNCs, GO and CNCs-GO within a certain time.

## 2.4. Adsorption of antibiotic

### 2.4.1. Adsorption of antibiotic

Levo-HCl solution with an initial concentration of $5–12 \text{ mg l}^{-1}$ and CNCs-GO composite adsorption material (0.025–0.125 g) were introduced into a 50 ml reaction bottle for batch adsorption experiments.

**Table 2.** Experimental independent variables and their levels.

| factor | coding | values for each level of coding | | |
|---|---|---|---|---|
| | | −1 | 0 | +1 |
| initial pollutant concentration (mg l$^{-1}$) | A | 9 | 10 | 11 |
| pH | B | 2 | 4 | 6 |
| dosage of adsorbent (g l$^{-1}$) | C | 0.5 | 1.0 | 1.5 |
| time (h) | D | 3 | 4 | 5 |

The pH was adjusted between 2 and 9 by adding different concentrations of NaOH and HCl solutions. The configured suspension was shaken for 6 h, with samples withdrawn at specific intervals and centrifuged at 10 000 r.p.m. for 15 min to separate the solid and liquid phases. The supernatant was removed and the concentration of Levo-HCl in the liquid phase was measured by HPLC (Agilent-1200, USA). The residual Levo-HCl concentration was obtained using a standard curve derived from a series of Levo-HCl solutions with known antibiotic content. The adsorbed amount of Levo-HCl was calculated according to equation (2.1),

$$q_t = \frac{(C_0 - C_t) \times V}{m},$$
(2.1)

where $q_t$ is the amount adsorbed after time $t$, $C_0$ and $C_t$ are initial concentration and concentration of the adsorbate after time $t$, respectively (mg l$^{-1}$); $V$ is the volume of the solution (l) and $m$ is the weight of the CNCs-GO used (g).

The percentage removal ($R\%$) of the Levo-HCl was calculated using equation (2.2):

$$R\% = \frac{(C_0 - C_t)}{C_0} \times 100\%.$$
(2.2)

### 2.4.2. Statistical analysis and response surface design

Data analysis was performed using IBM SPSS Statistics 19.0, USA and Stat-Ease Design-Expert 10.0, USA. According to RSM design principles, the Box–Behnken model was used to study four factors impacting Levo-HCl removal from water by the CNCs-GO: initial pollutant concentration, initial pH of the solution, contact time and amount of adsorbent. We predicted the optimal conditions in the experimental range; the centre point experiment is three parallel experiments. The relationship between the four factor, three horizontal coding and experimental values is shown in table 2.

# 3. Results and discussion

## 3.1. Characterization of the adsorbent

### 3.1.1. Scanning electron microscope–energy-dispersive spectroscopy and transmission electron microscopy analysis

Figure 1a,b presents SEM images of the CNCs at 5000× and 30 000× magnifications, respectively. Many rod-like cellulose crystals are observed on the CNCs surfaces, probably due to the large number of hydroxyl groups they contain. After hydrolysis treatment, the size of cellulose becomes smaller and the specific surface area increases. The hydroxyl groups on CNCs surface are more likely to interact with each other to form hydrogen bonds, and the irregular surface roughness further increases the surface area [35,36]. In figure 1c,d are the 5000× and 30 000× SEM images for GO, respectively. Here, the GO sheet layers overlap to form a three-dimensional structure. The GO sheet layers have many folds stacked on the surface, which account for its relatively large specific surface area. These folds are potential adsorption sites. The analogous SEM images for the CNCs-GO are presented in figure 1e,f. In this case, the surface morphology of the composite material reveals the appearance of CNCs on the surface of the three-dimensional GO structure, resulting in an even larger specific surface area and a greater number of adsorption sites.

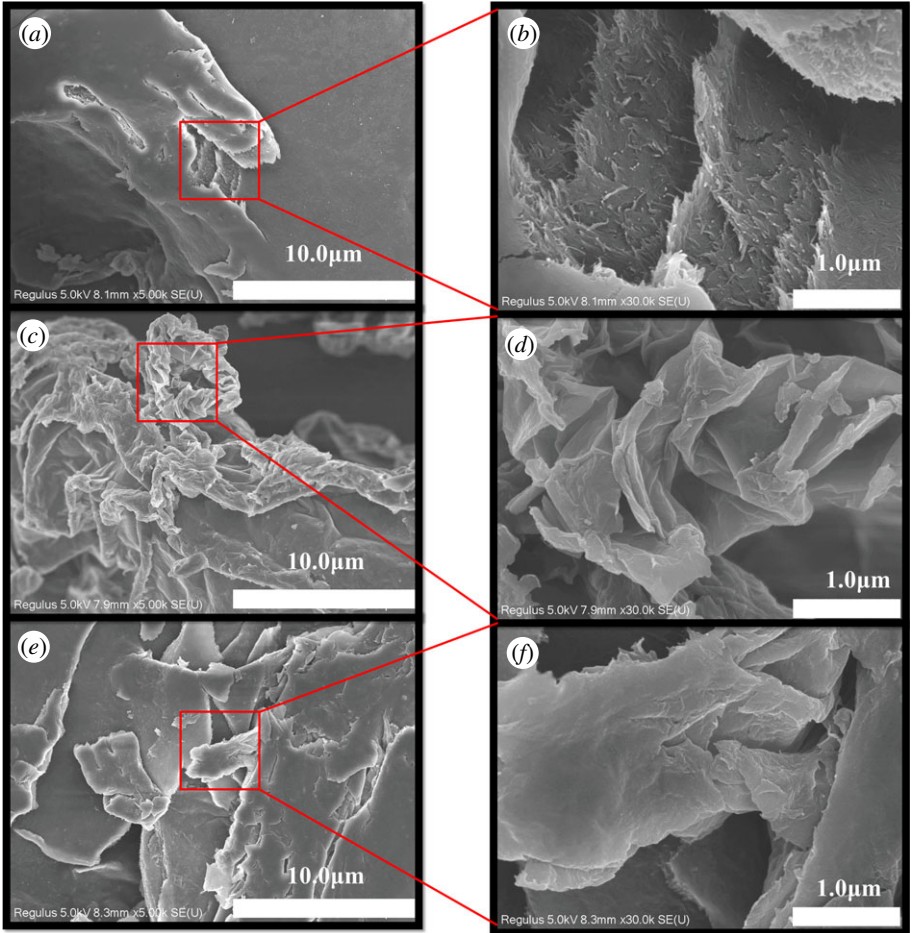

**Figure 1.** SEM images of CNCs, GO and CNCs-GO: (*a*) SEM images of CNCs at 5000× magnification; (*b*) SEM images of CNCs at 30 000× magnification; (*c*) SEM images of GO at 5000× magnification; (*d*) SEM images of GO at 30 000× magnification; (*e*) SEM images of CNCs-GO at 5000× magnification; (*f*) SEM images of CNCs-GO at 30 000× magnification.

Figure 2 demonstrates the transmission electron microscopy (TEM) micrographs of CNCs-GO. From the TEM micrograph, it can be seen that nano-sized CNCs and GOs CNCs-GO composites are formed. The CNCs are evenly distributed between the GO slices, interlaced with each other. At the same time, there are fewer folds of GO sheets, which may be due to the interaction between CNCs and GO, which prevents the folding of GO sheets. These data provide a basis for the successful preparation of composite materials.

The EDS spectra of the CNCs, GO and CNCs-GO, presented in figure 3, show that the CNCs and GO contain mostly C and O atoms, and with a small number of S atoms. Unmarked peaks due to other elements may be due to adventitious impurities not removed during the preparation and washing processes. Comparing the mass and element ratios, the CNCs-GO has both a larger C content than GO and a greater O content than the CNCs, indicating that the numbers of carbon and oxygen atoms in the composite material are increased (because the CNCs contain more carbon atoms and GO contains more oxygen atoms). Thus, the EDS analysis also provides clear support for successful composite formation.

### 3.1.2. Dynamic light scattering analysis

Dynamic light scattering (DLS) measures particle size on the basis of fluctuations in scattered light intensity with time that may be due to the random Brownian motion of the sample particles present in suspension or polymers in a solution. Diffusion is directly related to the statistical nature of these fluctuations in scattered intensity [37,38]. In addition, dynamic light scattering can be used to help prove whether the prepared composite material is at the nano-scale level [39]. As shown in figure 4, the CNCs-GO were unimodal in distribution, all the particles are in a range between 300 and 1000 nm, and the mean average size of the resulting nanoparticles was found to be 842.3 (nm),

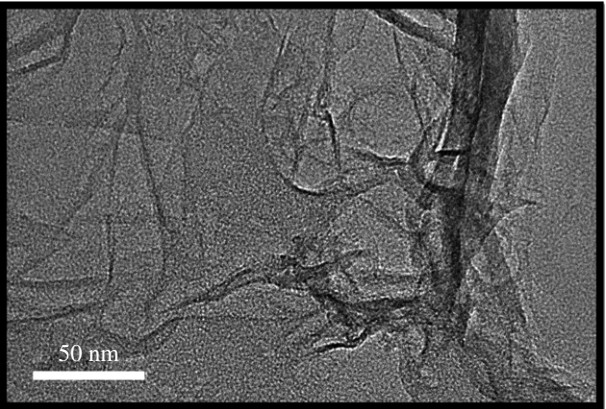

**Figure 2.** TEM images of CNCs-GO.

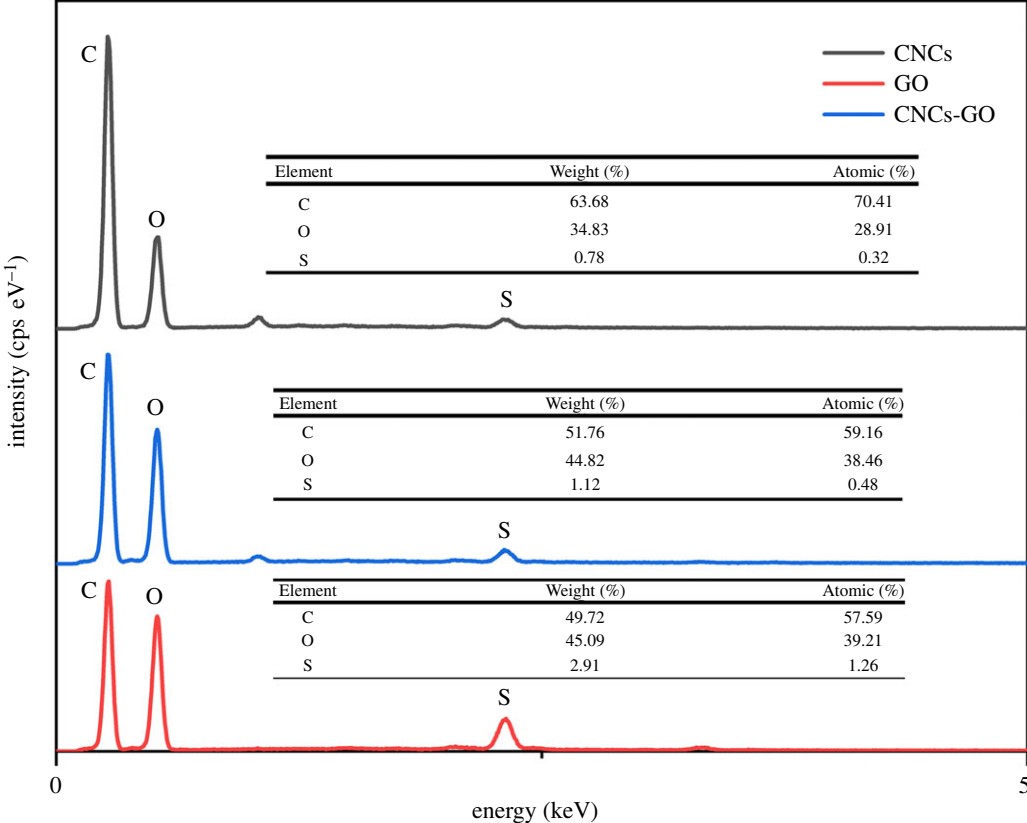

**Figure 3.** EDS spectra of CNCs, GO and CNCs-GO.

showing highly stable without forming any aggregation. At the same time, a narrow size distribution indicates many particles that are homogeneous in size with small variations; this result proves that the composite particles have a nano-size.

### 3.1.3. Fourier transform infrared analysis

The FT-IR patterns of the CNCs, GO and CNCs-GO are depicted in figure 5. The GO spectrum reveals absorption peaks at 3340, 1732, 1621, 1218 and 1049 cm$^{-1}$, corresponding to –OH stretching motions, C=O stretching vibrations in –COOH groups, C=C stretching vibrations of $sp^2$ hybridized carbon chains, C–OH stretching vibrations in –COOH groups and C–O–C stretching vibrations. The absorption peak at 864 cm$^{-1}$ is due to C–H bending vibrations. In the IR spectrum of the CNCs, absorption peaks are observed at 3340, 2900, 1649, 1428 and 1058 cm$^{-1}$, which correspond to –OH

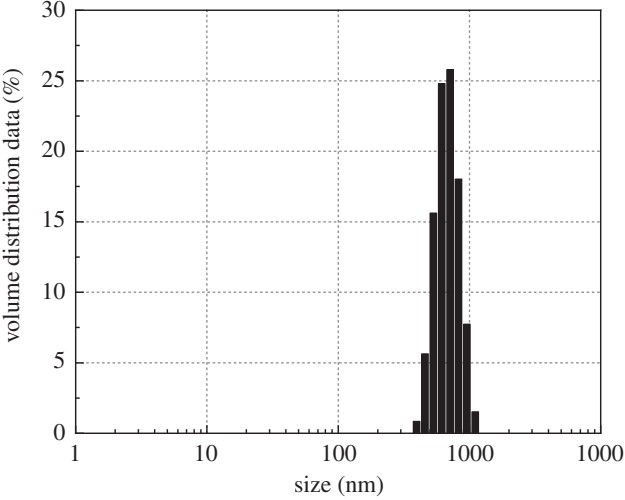

**Figure 4.** Spectroscopic measurement of particle size by dynamic light scattering of CNCs-GO.

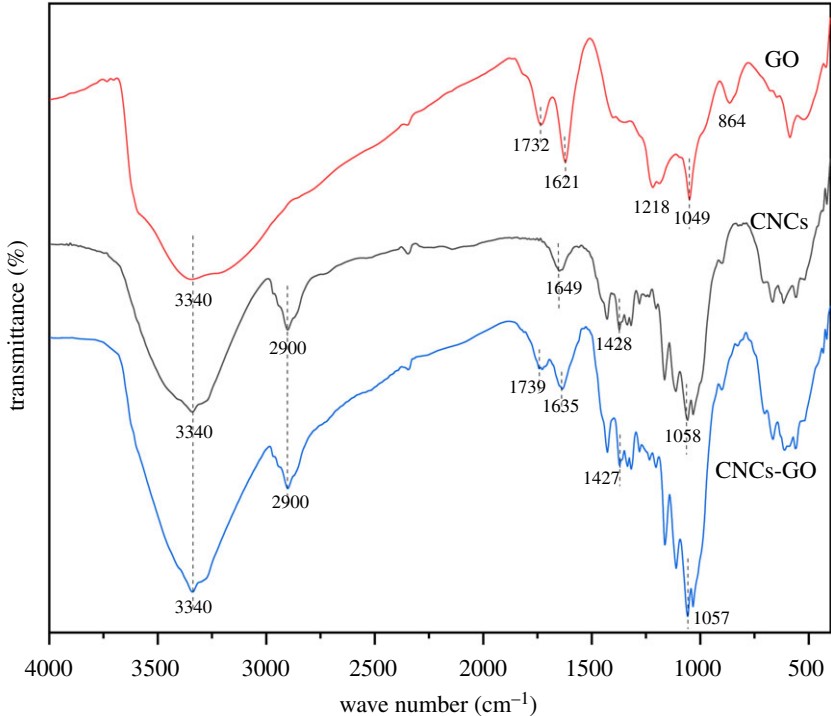

**Figure 5.** FT-IR spectra of GO, CNCs and CNCs-GO.

stretching vibrations, C–H stretching vibrations, OH bending vibrations and C-OH stretching vibration and C-O stretching vibration in carboxyl group. The FT-IR spectrum of the CNCs-GO composite contains all the absorption peaks observed in GO, as well as peaks unique to nanocellulose, such as that at 2900 cm$^{-1}$, corresponding to the –C–H stretching vibrations of its methyl and methylene groups. These FT-IR results also affirm that the CNCs-GO composite was successfully prepared.

### 3.1.4. Pore specifications and surface area analysis

$N_2$ adsorption–desorption isotherms and Barrett–Joyner–Halenda (BJH) pore volume distributions for the CNCs and CNCs-GO are shown in figure 6. The CNCs produce type II isotherms, in contrast with the CNCs-GO, which produce type IV isotherms. In the low-pressure range of $P/P_0 < 0.1$, the $N_2$ adsorption–desorption curves of the CNCs and CNCs-GO rise sharply, indicating that both materials have a certain number of micropores. In the relative pressure range of 0.4–0.8, there is a significant

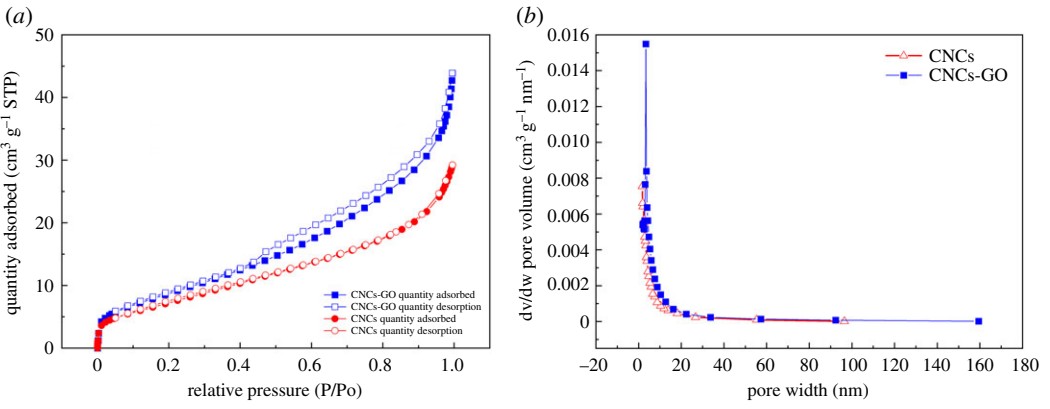

**Figure 6.** N$_2$ adsorption–desorption isotherms and BJH pore volume distributions of CNCs and CNCs-GO: (*a*) N$_2$ adsorption–desorption isotherms of CNCs and CNCs-GO, (*b*) Pore volume distribution of CNCs and CNCs-GO.

**Table 3.** Specific surface area and pore structure parameters of CNCs and CNCs-GO.

| sample | BET (m$^2$ g$^{-1}$) | BJH adsorption cumulative volume of pores (cm$^3$ g$^{-1}$) | single point surface area (m$^2$ g$^{-1}$) | adsorption average pore diameter (nm) |
|---|---|---|---|---|
| CNCs | 27.2370 | 0.043409 | 26.6827 | 6.37498 |
| CNCs-GO | 32.7995 | 0.062377 | 32.0515 | 7.60709 |

hysteresis loop in the N$_2$ adsorption–desorption curve of the CNCs-GO, indicating the existence of mesoporous structure. In the relative pressure range of 0.8–1.0, the curve of the CNCs-GO increases significantly, indicating that there are large cavities in the composite. The N$_2$ adsorption–desorption curve of the CNCs has no obvious hysteresis loop in the 0.4–1.0 relative pressure range, indicating that this material has no mesoporous or hollow structures. Figure 6*b* presents the pore size distribution curves for the CNCs and CNCs-GO calculated using the BJH model. Detailed structural parameters for the different samples are shown in table 3. As expected, the CNCs-GO has the largest BET surface area and pore volume. These results indicate CNCs-GO compound material with ultrahigh surface areas might be effective in adsorbing antibiotics.

### 3.1.5. X-ray diffraction analysis

The XRD patterns for the CNCs, GO and CNCs-GO are shown in figure 7. The CNCs display three distinct characteristic peaks at $2\theta = 14.9°$, 16.5° and 22.8°, indicating that they possess the typical monoclinic cellulose *I* lattice [40]. The concave area at $2\theta = 17.8°$ corresponds to an amorphous area of the CNCs. Compared to the crystalline regions, the amorphous areas of the CNCs afford higher chemical reactivity. In the CNCs-GO, the peak at 22.5° confirms the amorphous nature of the composite. GO has a characteristic peak at $2\theta = 9.9°$ which corresponds to the (001) crystal plane of GO; this arises from the lattice distortion of the carbon structure during the transformation of graphite to GO, during which the oxidation process introduces a large number of functional groups and the interlayer spacing is enlarged, so that the graphite layers are separated and the specific surface area and the number of adsorption sites are increased [41,42]. The characteristic peaks of the CNCs-GO, observed at $2\theta = 15.1°$, 16.2° and 22.6°, are similar to those of the CNCs, and show that the composite generates almost the same spectrum as the CNCs after slight modification. Further, the peak occurring at $2\theta = 10.21°$ indicates that the composite material also retains the characteristics of GO. Therefore, it can be proven that the CNCs-GO composite is a substance comprising CNCs and GO.

### 3.1.6. X-ray photoelectron spectroscopy

The XPS analysis results for the CNCs are displayed in figure 8*a,d,g*. The full spectrum scan shows that the main chemical components of the CNCs are C (C 1 *s*, 287 eV) and O (O 1 *s*, 533 eV). The C 1 *s*

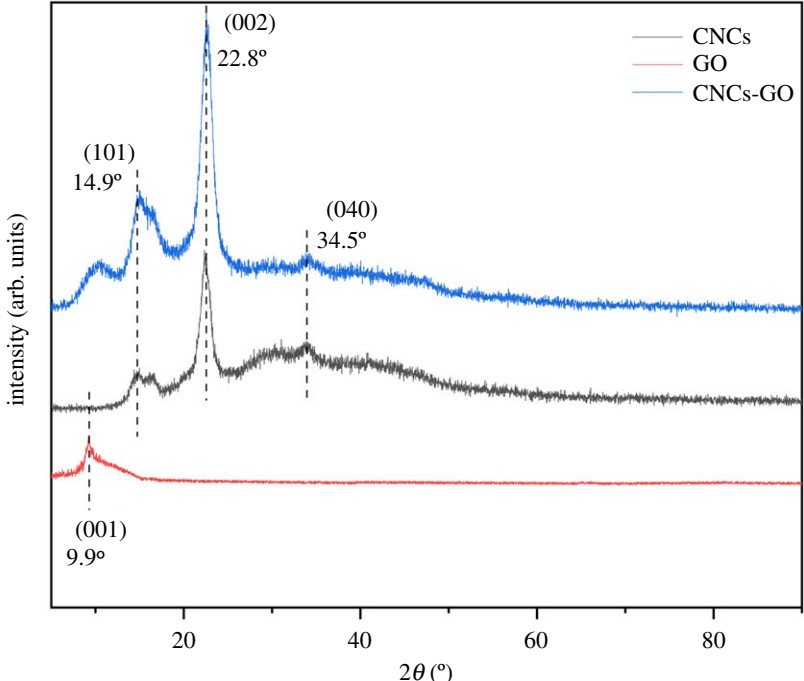

**Figure 7.** X-ray diffraction patterns of CNCs, GO and CNCs-GO.

component peak scan shows that C is mainly present in three forms: C–C/C–H, C–O and O–C=O, with corresponding peaks at 284.8, 286.2 and 287.5 eV, respectively. The O 1 s spectrum reveals that O is mainly present in the forms C–O and C=O, with peaks at 532.5 and 531.7 eV, respectively. The XPS analysis results for GO are shown in figure 8b,e,h. The full spectrum scan reveals that the main chemical components are C (C 1 s, 285 eV) and O (O 1 s, 533.6 eV). The C 1 s spectrum is decomposed into peaks for four carbon forms: C–C/C=C, C–O, C=O and O–C=O, corresponding to peaks at 284.8, 286.7, 287.2 and 288.5 eV, respectively, whereas the O 1 s deconvolution shows the presence of three types of O, C–O, C=O and O–C=O (at 532.6, 531.7, and 532.8 eV, respectively). The XPS analysis results for the CNCs-GO composite are shown in figure 8c,h,i. In the full spectrum scan, the main chemical components of the CNCs-GO are also C (C 1 s, 287 eV) and O (O 1 s, 533 eV) [43]. The C 1 s peak decomposition reveals four peaks for C at 284.9, 285.4, 286.2 and 287.2 eV, corresponding to C–O, C–C/C=C, O–C=O and C=O functional groups. Compared with that in the scan for CNCs, the C–C/C=C peak intensity in the scan for CNCs-GO increases significantly, indicating that the CNCs and GO are effectively combined. For the C 1 s spectrum of the CNCs, the addition of GO introduces –C=O groups in the composite [44]. This should be advantageous for the adsorption of antibiotics by the new adsorbent [45]. In addition, here still remain a large number of oxygen-containing functional groups in the CNCs-GO, mainly derived from the CNCs. The O 1 s peak separation results show three different peaks for O at 532.1, 532.6 and 533.1 eV, corresponding to the functional groups C=O, C–O and O–C=O, respectively, which are typically found in GO. Compared with that in the scan for CNCs, the O peak intensity in the scan for CNCs-GO is significantly increased, indicating that GO effectively binds to the CNCs through hydrogen bonding. For the CNCs-GO, both the O 1 s and C 1 s spectra exhibit changes consistent with the formation of the composite material, with respect to the corresponding spectra of the CNCs.

### 3.1.7. Contact angle analysis

The hydrophilicity characteristics of the CNCs, GO and CNCs-GO composites were analysed by contact angle measurements. Figure 9a shows a contact angle for GO of 33.93°, indicating good hydrophilicity [46]. In the nanocellulose (figure 9b), the contact angle is 23.63°, also signalling good hydrophilicity. Finally, the 26.72° contact angle of the CNCs-GO composite figure 9c similarly suggests good hydrophilicity. With nanocellulose as the matrix and the GO as filler, the contact angle of the

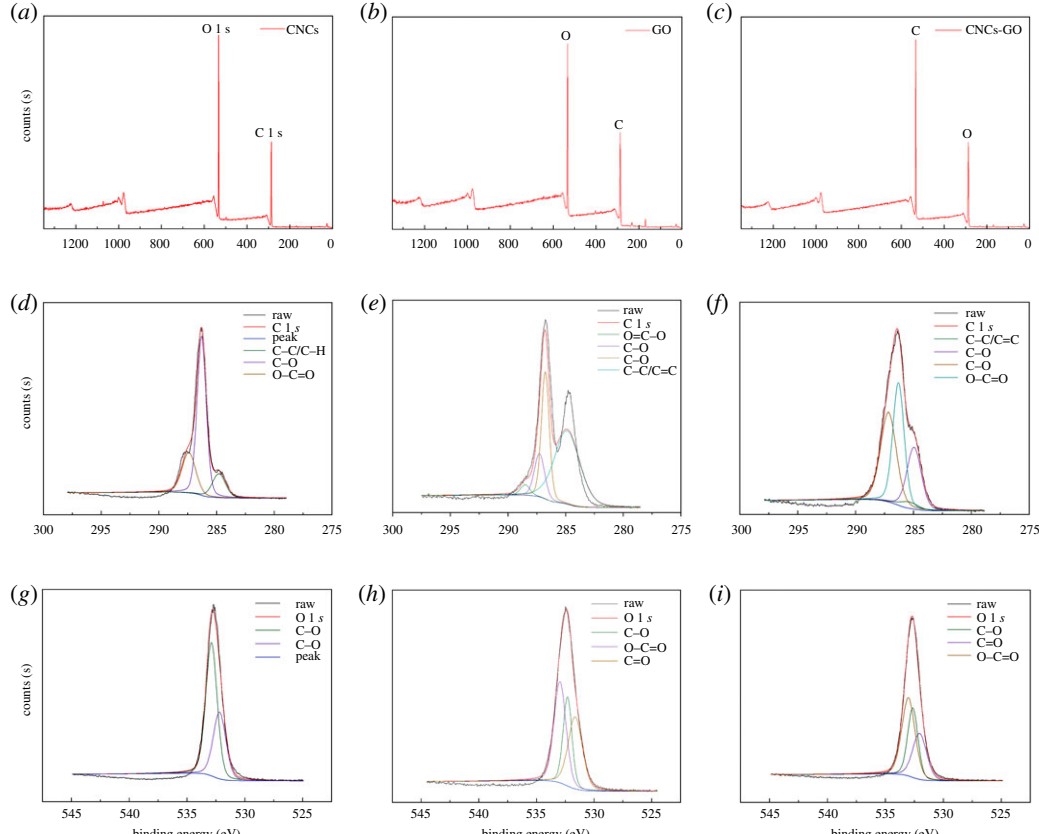

**Figure 8.** X-ray photoelectron spectroscopy of CNCs, GO and CNCs-GO.

composite was reduced and the hydrophilicity was improved. The contact angle analysis indicates the successful preparation of the CNCs-GO material.

## 3.2. Response surface analysis

### 3.2.1. Response surface experiment design and results

The statistical software Design-Expert 10.0 was used to fit the multiple regression equations to the experimental data from the response surface in table 4 to obtain the initial pollutant concentration ($A$), pH ($B$), amount of adsorbent added ($C$), contact time ($D$) and the quadratic polynomial equation for the Levo-HCl removal rate ($\eta$), as shown in equation (3.1),

$$\eta = 82.64 + 14.64A + 15.40B + 5.09C + 1.29D + 10.68AB - 1.68AC$$
$$+ 0.82AD - 2.35BC - 3.96BD - 0.47CD - 19.07A^2 - 15.66B^2 - 6.87C^2 - 2.72D^2. \tag{3.1}$$

### 3.2.2. Analysis of regression model variance and principal factors

The results from the regression analysis of variance are shown in table 5. Among the $p$-values, if an item is $p < 0.05$, the effect of the item on the response value is significant; if a certain item is $p < 0.01$, the effect of the item on the response value is very significant [32]. The smaller the lack of fit of the model, the better, and the larger the $p$-value corresponding to the lack of fit, the better. The larger the $p$-value, the better the $p$-value. From table 5, the independent variables $A$, $B$, $C$, $AB$, $A^2$, $B^2$ and $C^2$ respond significantly ($p < 0.05$), that is, the initial pollutant concentration, the initial pH, the amount of adsorbent added, the interaction between initial pH and initial pollutant concentration, the square of pollutant concentration, the square of the initial pH and the square of the amount of adsorbent added all have significant effects, and other factors have little effect on the response value $\eta$. The adaptability of the model is extremely significant ($p < 0.01$), and the misfit term is not significant ($p = 0.0534$), indicating that the model can well describe the nonlinear relationship between the influencing factors and the response value [47]. The model's coefficient of determination ($R^2 = 0.9494$) can explain 94.94%

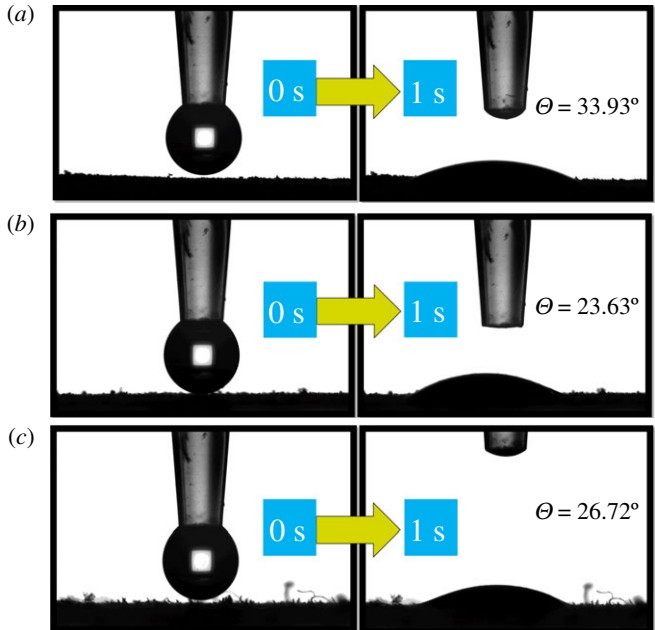

**Figure 9.** Contact angle diagrams of GO (*a*), CNCs (*b*) and CNCs-GO (*c*).

of the experimental data [48]. The difference between the corrected complex correlation coefficient ($R^2_{Adj}$) and the predicted complex correlation coefficient ($R^2_{pred}$) is 0.1764, which is less than 0.2; the coefficient of variation CV = 9.46%, and the precision is 13.738, which is much greater than 4.0, indicating that the model has high precision and reliability [32,49]. It further shows that this experimental method is reliable. Therefore, the model can be used to optimize and predict the experimental conditions for the removal of Levo-HCl by the CNCs-GO in water.

### 3.2.3. Response surface analysis of levofloxacin hydrochloride removal rate

By drawing three-dimensional images, each with two independent variables as coordinates, using Stat-Ease Design-Expert 10.0, we can more intuitively explain the effects of sorbent dosage, initial pollutant concentration, solution pH and contact time on the removal rate, and characterize the response surface function. The results are shown in figures 10 and 11.

Figure 10*a* shows the influence of the solution pH and initial pollutant concentration on the removal rate under the centre point conditions of adsorbent dosage (1.0 g l$^{-1}$) and contact time (4 h). Figure 10*a* shows that, as the initial pH and pollutant concentration increase, the removal rate also gradually increases. With an initial pH of 4 and initial pollutant concentration of 10.0 mg l$^{-1}$, the removal rate reaches a maximum of 80.12%. This is due to the increase in the initial concentration of the pollutant: providing a sufficient number of antibiotic molecules for the adsorbent's adsorption sites effectively results in a higher pollutant concentration, and therefore, a higher removal rate. From the contour map, the interaction between the initial concentration and pH lies close to the ellipse, so there is an obvious interaction between the initial concentration and the initial pH of the solution.

Figure 10*b* shows the effects of the adsorbent dosage and initial pollutant concentration on the removal rate for an initial pH and contact time under the centre point conditions (pH 4; contact time, 4 h). As the initial pollutant concentration and amount of adsorbent added increase, the removal rate also gradually increases. When the initial concentration of the pollutant is 10.0 mg l$^{-1}$ and the adsorbent dosage is 1.0 g l$^{-1}$, the removal rate reaches a maximum of 80.12%. Here, as the amount of adsorbent added increases, sufficient adsorption sites are provided to adsorb antibiotic molecules, resulting in a higher removal rate. When the initial concentration is greater than 10.0 mg l$^{-1}$ and the adsorbent dose exceeds 1.0 g l$^{-1}$, the removal rate tends to slow. In this case, when adsorption reaches saturation, increasing the dosage will cause violent collisions between adsorbent particles, reduce the number of channels contacting with antibiotic molecules and thus affect the removal rate [50]. From the contour map, the shape of the interaction between the initial concentration and amount of adsorbent added is closer to an oval, indicating a clear interaction between the two conditions.

**Table 4.** Experimental design and results.

| numbering | variable coding level | | | | actual value | | | | removal rate (%) |
|---|---|---|---|---|---|---|---|---|---|
| | A | B | C | D | A | B | C | D | |
| 1 | 0 | −1 | −1 | 0 | 10 | 2 | 0.5 | 4 | 38.22 |
| 2 | 0 | 0 | 1 | −1 | 10 | 4 | 1.5 | 3 | 79.45 |
| 3 | −1 | 0 | −1 | 0 | 9 | 4 | 0.5 | 4 | 34.56 |
| 4 | 0 | 0 | 1 | 1 | 10 | 4 | 1.5 | 5 | 78.46 |
| 5 | 0 | 0 | −1 | 1 | 10 | 4 | 0.5 | 5 | 61.56 |
| 6 | 1 | 0 | −1 | 0 | 11 | 4 | 0.5 | 4 | 74.56 |
| 7 | −1 | −1 | 0 | 0 | 9 | 2 | 1.0 | 4 | 31.57 |
| 8 | 0 | −1 | 0 | 1 | 10 | 2 | 1.0 | 5 | 60.57 |
| 9 | 0 | 0 | 0 | 0 | 10 | 4 | 1.0 | 4 | 80.57 |
| 10 | 0 | 0 | 0 | 0 | 10 | 4 | 1.0 | 4 | 80.93 |
| 11 | 0 | −1 | 1 | 0 | 10 | 2 | 1.5 | 4 | 47.55 |
| 12 | 1 | 0 | 1 | 0 | 11 | 4 | 1.5 | 4 | 79.25 |
| 13 | 1 | 0 | 0 | 1 | 11 | 4 | 1.0 | 5 | 79.76 |
| 14 | 0 | 0 | 0 | 0 | 10 | 4 | 1.0 | 4 | 80.12 |
| 15 | 1 | 0 | 0 | −1 | 11 | 4 | 1.0 | 3 | 77.17 |
| 16 | −1 | 0 | 0 | 1 | 9 | 4 | 1.0 | 5 | 45.16 |
| 17 | 0 | 1 | 0 | −1 | 10 | 6 | 1.0 | 3 | 79.63 |
| 18 | −1 | 0 | 1 | 0 | 9 | 4 | 1.5 | 4 | 45.96 |
| 19 | 1 | 1 | 0 | 0 | 11 | 6 | 1.0 | 4 | 79.56 |
| 20 | 0 | 1 | 0 | 1 | 10 | 6 | 1.0 | 5 | 78.55 |
| 21 | 0 | 1 | 1 | 0 | 10 | 6 | 1.5 | 4 | 79.56 |
| 22 | 1 | −1 | 0 | 0 | 11 | 2 | 1.0 | 4 | 28.42 |
| 23 | 0 | 0 | 0 | 0 | 10 | 4 | 1.0 | 4 | 85.35 |
| 24 | 0 | 1 | −1 | 0 | 10 | 6 | 0.5 | 4 | 79.73 |
| 25 | 0 | −1 | 0 | −1 | 10 | 2 | 1.0 | 3 | 45.82 |
| 26 | 0 | 0 | −1 | −1 | 10 | 4 | 0.5 | 3 | 60.66 |
| 27 | −1 | 0 | 0 | −1 | 9 | 4 | 1.0 | 3 | 45.86 |
| 28 | −1 | 1 | 0 | 0 | 9 | 6 | 1.0 | 4 | 39.98 |
| 29 | 0 | 0 | 0 | 0 | 10 | 4 | 1.0 | 4 | 86.22 |

Figure 10$c$ maps the effects of contact time and initial pollutant concentration on the removal rate based on the solution pH and the amount of adsorbent added under the centre point conditions (pH 4; adsorbent dose, 1.0 g l$^{-1}$). When the contact time and initial pollutant concentration increase, the removal rate also gradually increases. When the initial concentration of the pollutant is 10.0 mg l$^{-1}$ and the contact time is 4 h, the removal rate reaches a maximum of 80.12%, because when the contact time is increased, adsorption sites have more time for contact with and adsorption of the antibiotic molecules, resulting in a higher removal rate. When the maximum removal rate is reached, as the influencing factors increase, the removal rate begins to plateau, as adsorption equilibrium is achieved. The contour map reveals that the interaction between the contact time and initial pollutant concentration is closer to an oval, so there is a clear interaction between the contact time and the initial concentration of the pollutant.

The influence of the adsorbent concentration and initial solution pH on the removal rate under the centre point conditions of the pollutant concentration (10.0 mg l$^{-1}$) and contact time (4 h) is shown in figure 11$a$. When the amount of adsorbent and initial pH are increased, the removal rate gradually

**Table 5.** Variance analysis results of regression model. $R^2 = 0.9494$, $R^2_{Adj} = 0.8989$, $R^2_{pred} = 0.7225$, CV = 9.46%, adequate precision = 13.738.

| source | sum of squares | df | mean square | F value | p-value Prob > F |
|---|---|---|---|---|---|
| model | 9714.15 | 14 | 693.87 | 18.77 | <0.0001 |
| A | 2570.49 | 1 | 2570.49 | 69.55 | <0.0001 |
| B | 2844.69 | 1 | 2844.69 | 76.97 | <0.0001 |
| C | 310.49 | 1 | 310.49 | 8.40 | 0.0117 |
| D | 19.94 | 1 | 19.94 | 0.54 | 0.4747 |
| AB | 456.46 | 1 | 456.46 | 12.35 | 0.0034 |
| AC | 11.26 | 1 | 11.26 | 0.30 | 0.5897 |
| AD | 2.71 | 1 | 2.71 | 0.073 | 0.7907 |
| BC | 22.09 | 1 | 22.09 | 0.60 | 0.4523 |
| BD | 62.65 | 1 | 62.65 | 1.70 | 0.2140 |
| CD | 0.89 | 1 | 0.89 | 0.024 | 0.8787 |
| $A^2$ | 2359.07 | 1 | 2359.07 | 63.83 | <0.0001 |
| $B^2$ | 1591.61 | 1 | 1591.61 | 43.06 | <0.0001 |
| $C^2$ | 306.09 | 1 | 306.09 | 8.28 | 0.0122 |
| $D^2$ | 47.84 | 1 | 47.84 | 1.29 | 0.2744 |
| residual | 517.44 | 14 | 36.96 | | |
| lack of fit | 483.72 | 10 | 48.37 | 5.74 | 0.0534 |
| pure error | 33.72 | 4 | 8.43 | | |
| corrected total | 10 231.59 | 28 | | | |

increases. For an adsorbent dosage of 1.0 g l$^{-1}$ and initial pH of 4, the removal rate reaches a maximum of 80.12%. In this case, the higher adsorbent dosage causes an increase in the number of adsorption sites. At the same time, when the pH is in the lower range, many positive charges are formed on the surface of the remaining adsorbent; the higher proportion of adsorption sites with positive charges, which strongly and electrostatically attract the antibiotic molecules, produces a higher removal rate. As the pH increases, the negative charge content on the surface of the adsorbent increases, decreasing its electrostatic attractions with the antibiotic molecules and decreasing the removal rate. The contour map reveals that the interaction between the dosage of the adsorbent and initial pH is closer to an oval, indicating the clear interaction between the dosage of the adsorbent and initial pH.

The influence of contact time and initial solution pH on the removal rate under the centre point conditions of pollutant concentration (10.0 mg l$^{-1}$) and adsorbent dosage (1.0 g l$^{-1}$) is depicted in figure 11b. When the contact time and initial pH increase, the removal rate also gradually increases. After 4 h at an initial pH of 4, the removal rate reaches a maximum of 80.12%; the increase in contact time enables more adsorption sites to contact the antibiotic molecules and adsorb them. However, as the pH rises, the removal rate gradually declines because the number of positive charges distributed on the surface of the remaining adsorbent decreases, resulting in weaker electrostatic attractions with the antibiotic molecules and a lower removal rate. From the contour map, the interaction between the contact time and initial solution pH is closer to an oval, denoting a clear interaction between these two parameters.

Finally, figure 11c presents the effect of contact time and the amount of adsorbent on the removal rate under the centre point conditions of the pollutant concentration (10.0 mg l$^{-1}$) and initial solution pH (pH 4). Accordingly, as the contact time and the amount of adsorbent added are increased, the removal rate is gradually increased. For a contact time of 4 h and adsorbent dosage of 1.0 g l$^{-1}$, the removal rate reaches a maximum of 80.12%. Here, the increases in the contact time and adsorbent dosage result in more adsorption sites having more time to contact and adsorb the antibiotic molecules, and hence, a higher removal rate. It can be seen from the contour map that the interaction

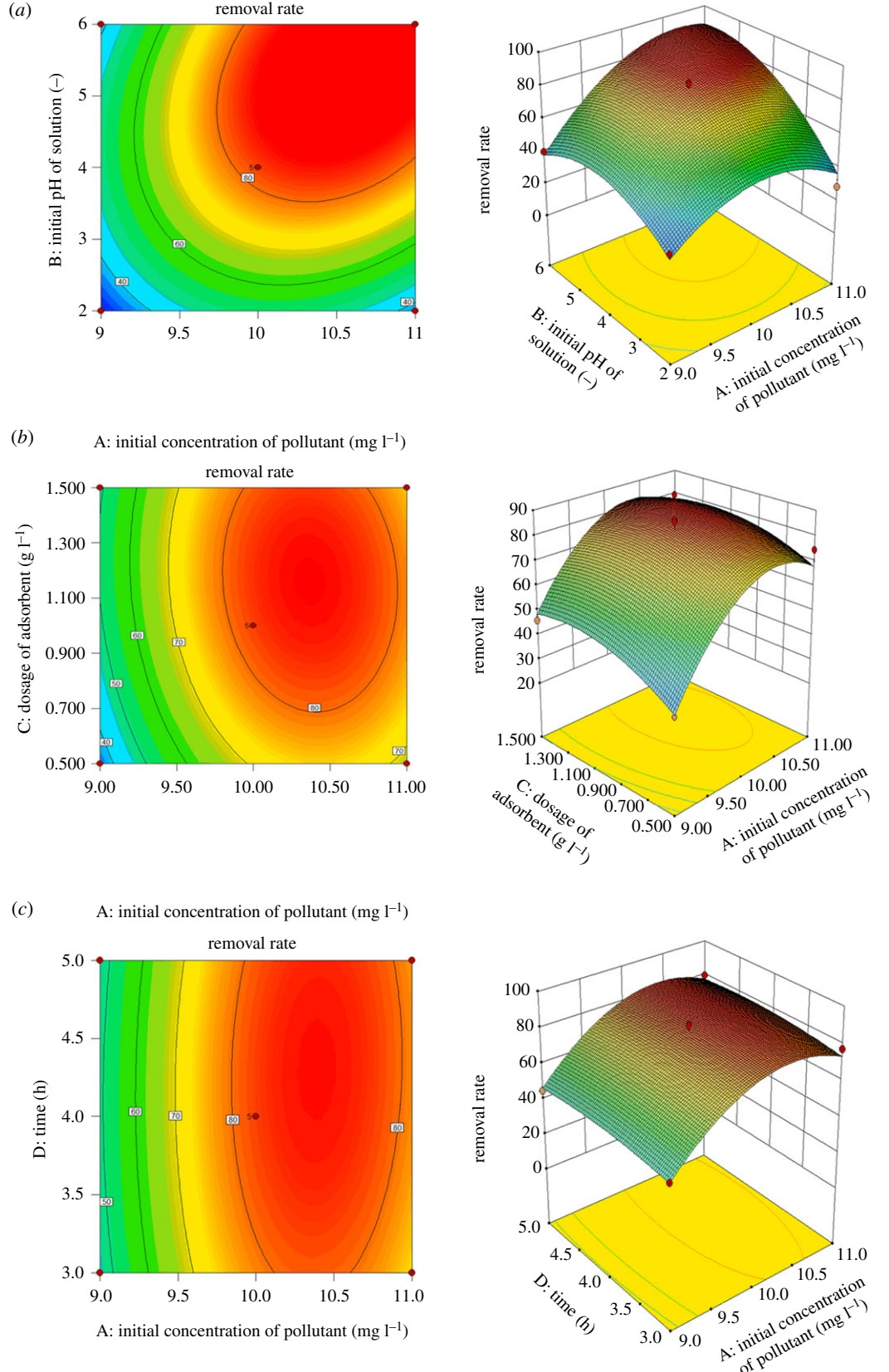

**Figure 10.** Surface and contour plots for Levo-HCl removal (%) interaction between independent parameters. (*a*) Initial pH of solution–Levo-HCl concentration. (*b*) the dosage of CNCs-GO–Levo-HCl concentration. (*c*) Time–Levo-HCl concentration.

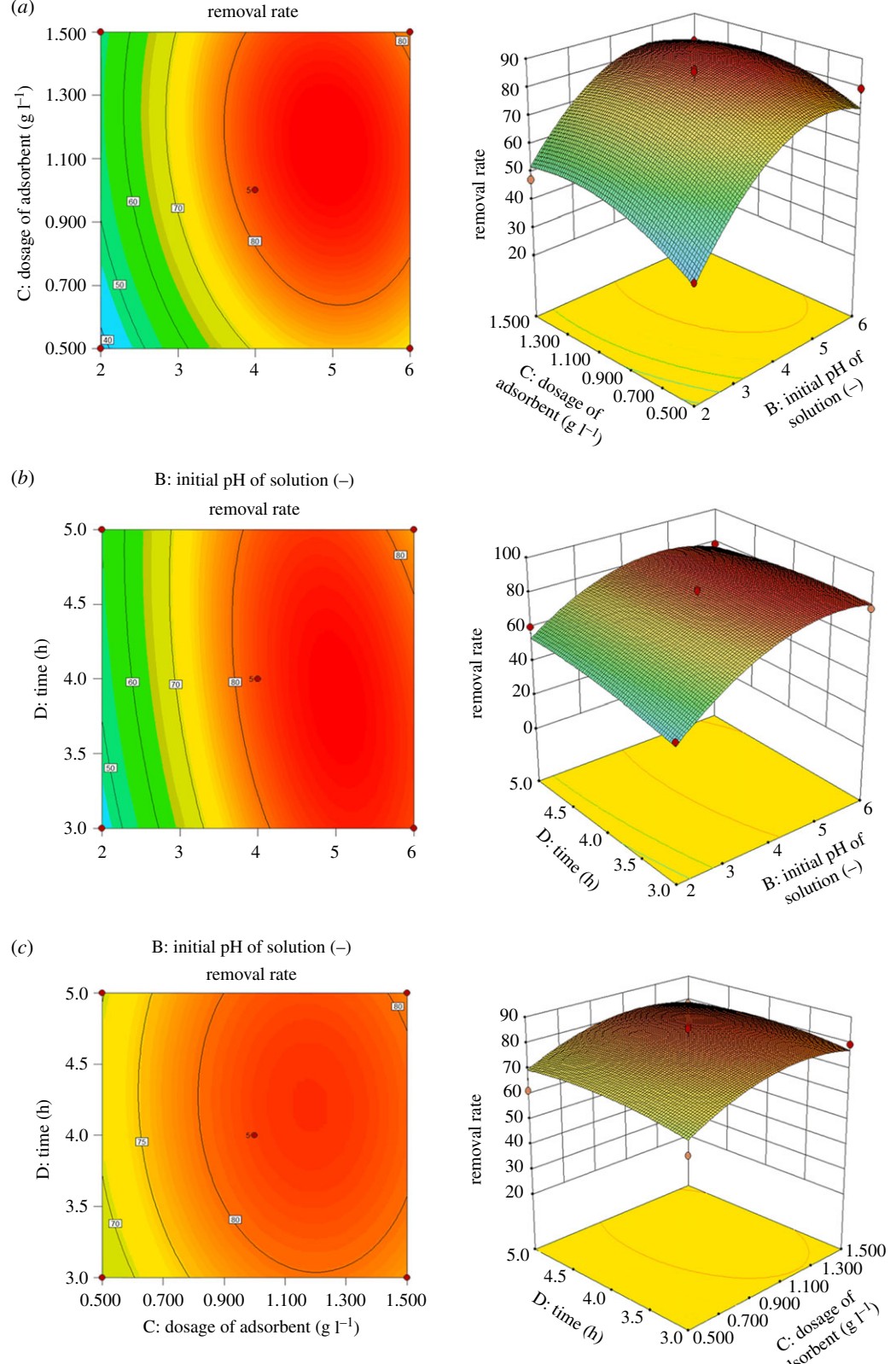

**Figure 11.** Surface and contour plots for Levo-HCl removal (%) interaction between independent parameters. (*a*) Initial pH of solution–the dosage of CNCs-GO. (*b*) Time–initial pH of solution. (*c*) Time–the dosage of CNCs-GO.

between the contact time and adsorbent dosage is closer to an oval, indicating a clear interaction between the contact time and adsorbent loading.

The Design-Expert 10.0 program was used to predict the optimal experimental conditions for the adsorption of Levo-HCl by the CNCs-GO, as follows: adsorbent dosage, $1.0\,g\,l^{-1}$; initial pollutant

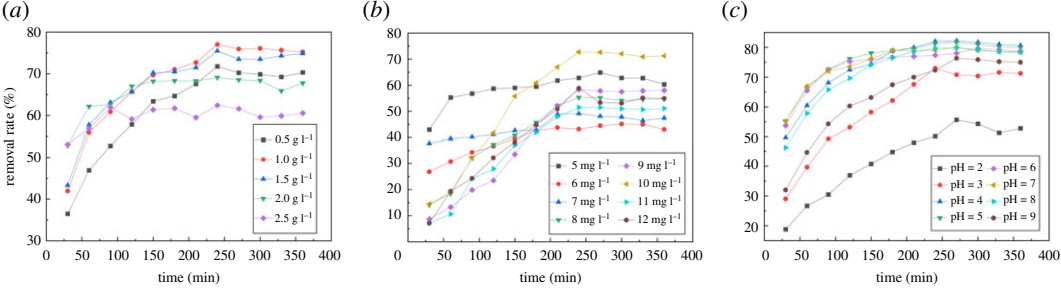

**Figure 12.** Adsorption of levofloxacin hydrochloride by CNCs-GO under different influence factors.

concentration, 10.0 mg l$^{-1}$; solution pH, 4; and contact time, 4 h. Under these conditions, the Levo-HCl removal rate can reach 80.12%. To verify the accuracy of the response surface model, three parallel experiments were performed under the predicted optimal conditions; the average value of the measured removal rate was 80.06%, indicating that the calculation model has good predictive ability.

The effect of the amount of adsorbent on the adsorption of Levo-HCl by the prepared composite is shown in figure 12a. The reaction reaches the adsorption equilibrium state at 240 min, and the removal rate reaches 77.01% at an adsorbent dosage of 1.0 g l$^{-1}$. The removal rate gradually decreases to 62.43% with the increase of adsorbent dosage from 1.0 to 2.5 g l$^{-1}$, which may be due to partial aggregation of the CNCs-GO at higher concentration and the concomitant reduction of effective adsorption sites [51].

Figure 12b shows the effect of the initial pollutant concentration on this adsorption system. As the initial concentration of antibiotic increases, its rate of removal by the CNCs-GO gradually increases, and the reaction reaches the adsorption equilibrium state at 240 min. For an initial concentration increase from 6.0 to 10.0 mg l$^{-1}$, the Levo-HCl removal rate increases from 44.45% to 72.59%. However, when the initial concentration rises from 10.0 to 12.0 mg l$^{-1}$, the removal rate decreases from 72.59% to 53.38%. Thus, within a certain concentration range, as the initial concentration of the pollutant increases, the probability of binding antibiotic molecules to the adsorption sites on the CNCs-GO surface increases, and the removal rate increases at a faster rate. However, when the initial concentration is higher, the limited adsorption sites and active sites tend to saturate, leading to reduced material removal rates.

The effect of the initial pH of the solution on the adsorption of Levo-HCl by the CNCs-GO is shown in figure 12c. The presence of H$^+$ or OH$^-$ in the solution changes the surface charge of the adsorbent. The figure reveals that the reaction reaches the adsorption equilibrium state at 240 min, and the removal rate is at least 55.68% at an initial pH of 2. As the pH increases, the removal rate gradually increases, reaching a maximum of 82.19% at pH 4. In the pH range 4–9, the removal rate gradually decreases. The variation of pH results in both the protonation–deprotonation of functional groups in the nanocellulose and graphene oxide and changes in the chemical speciation of ionizable organic compounds [52]. Levo-HCl can be adsorbed on the CNCs-GO by surface complexation or cation exchange. If surface complexation governs the sorption process, the adsorption capacity should not greatly decrease with the increase in solution pH. At lower pH values, the adsorption capacity is found to be reduced due to the decreased interaction between the antibiotic molecules and H$^+$ ions. However, as the pH increases, the adsorption capacity also increases slowly, due to the ionization of –COOH groups on the surface of the CNCs-GO. When the pH is higher than 4, the removal rate decreases. This enhances the formation of water clusters and lowers the formation of H-bonds [9]. However, H-bonds may contribute to the Levo-HCl/composite interaction in acidic medium. In addition, there are electrostatic interactions between the active material on the surface of the adsorbent and the lone pair electrons on the surface of Levo-HCl acid.

According to the response surface analysis of the second-order polynomial regression equation for the removal rate of Levo-HCl, the optimal adsorption conditions can be obtained. At the same time, single-factor experiments are performed to verify the results. The optimal conditions obtained by the two methods are the same, which proves that the model can better fit the actual values, and has certain guiding significance in practical application.

## 3.3. Adsorption kinetics

To clarify the adsorption mechanism and confirm which of three processes—chemical reaction, diffusion, or mass transfer—dominate in the rate-controlling step, the adsorption kinetics of the aqueous

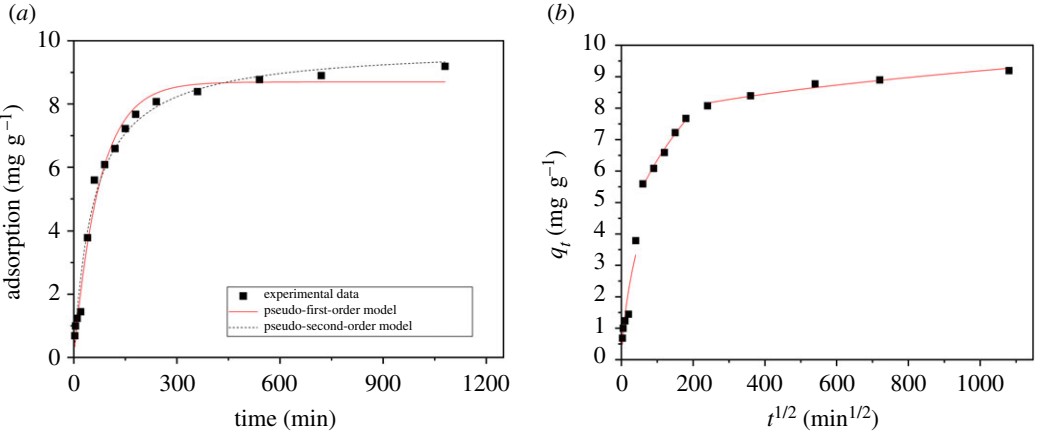

**Figure 13.** Adsorption kinetics of levofloxacin hydrochloride on CNCs-GO: (*a*) pseudo-first-order kinetic model, pseudo-second-order kinetic model; (*b*) intra-particle diffusion model.

Levo-HCl/CNCs-GO system was studied. The characteristics of the adsorption kinetics are shown in figure 13. The adsorption capacity gradually increases over time, exhibiting rapid adsorption in the initial stage (0–60 min) which can reach 60% of the equilibrium adsorption amount. The adsorption amount changes linearly with time, during which the surface-active sites of the CNCs-GO become occupied and the adsorption rate gradually decreases, eventually reaching saturation. It may be that, during the rapid adsorption phase, the dispersion of the CNCs-GO and Levo-HCl in water is high, and antibiotic molecules quickly diffuse to the surface-active sites of the adsorbent. During the slow adsorption phase, with the progress of the reaction, the Levo-HCl migrates toward the composite's internal pores and diffuses. This increases the mass transfer resistance, and the adsorption rate, affected by the nature of the composite material, solution pH and other factors, gradually decreases and finally reaches equilibrium.

To investigate the adsorption process of Levo-HCl on the CNCs-GO, pseudo-first- and pseudo-second-order kinetics models were used to fit the experimental results. The equations are expressed as follows equations (3.2) and (3.3):

$$\text{first-order kinetics model: } \ln(q_e - q_t) = \ln q_e - k_1 t \tag{3.2}$$

and

$$\text{second-order kinetics model: } \frac{t}{q_t} = \frac{1}{k_2 q_e^2} + \frac{t}{q_e}, \tag{3.3}$$

where $q_e$ and $q_t$ are the amounts of adsorbed Levo-HCl per unit mass of adsorbent (mg g$^{-1}$) at equilibrium and time $t$, respectively; $k_1$ is the pseudo-first-order sorption rate constant (min$^{-1}$); and $k_2$ is the rate constant of pseudo-second-order adsorption (g mg$^{-1}$ min$^{-1}$).

The kinetics parameters were calculated accordingly and are listed in table 6. Comparing the correlation coefficients of the two models, the fitting parameter $R^2$ of the pseudo-second-order kinetics equation is higher than that of the pseudo-first-order kinetics equation, indicating that the adsorption of Levo-HCl onto CNCs-GO does not follow pseudo-first-order kinetics. This may be due to control of the boundary layer in the initial stages of adsorption [53]. Therefore, a pseudo-second-order kinetics model is more suitable for describing the adsorption kinetics of Levo-HCl on CNCs-GO. Thus, the adsorption of Levo-HCl may be a process of chemical reaction to control the adsorption rate between Levo-HCl and CNCs-GO through electron sharing and electron exchange between particles [54]. The pseudo-first-order kinetics equation more suitably describes the initial stages of adsorption, with some limitations, while the pseudo-second-order kinetics equation reflects the complete adsorption process—membrane diffusion, surface adsorption and internal diffusion—and is more suitable for the current system [55]. The adsorption rate is directly proportional to the square of the pollutant concentration. The chemical reaction may be an important limiting factor for the adsorption of Levo-HCl by the CNCs-GO. This process may be affected by an interaction between the adsorbate and adsorbent or controlled by the exchange of electrons. The adsorption of Levo-HCl may be a process of chemical reaction to control the adsorption rate between Levo-HCl and CNCs-GO through electron sharing and electron exchange between particles.

**Table 6.** Parameters of kinetic model for levofloxacin hydrochloride adsorption onto CNCs-GO.

| kinetic model | | parameters | values |
|---|---|---|---|
| pseudo-first-order | | $q_e$ (mg g$^{-1}$) | 8.696 |
| | | $K_1$ (min$^{-1}$) | 0.0133 |
| | | $R^2$ | 0.9843 |
| pseudo-second-order | | $q_e$ (mg g$^{-1}$) | 9.7383 |
| | | $K_2$ (g mg$^{-1}$ min$^{-1}$) | 0.0017 |
| | | $R^2$ | 0.9867 |
| intra-particle diffusion | stage I | $K_{d1}$ (mg (g min$^{1/2}$)$^{-1}$) | 0.6201 |
| | | $C_1$ | -0.5926 |
| | | $R_1^2$ | 0.8149 |
| | stage II | $K_{d2}$ (mg (g min$^{1/2}$)$^{-1}$) | 0.3732 |
| | | $C_1$ | 2.6141 |
| | | $R_2^2$ | 0.9861 |
| | stage III | $K_{d3}$ (mg (g min$^{1/2}$)$^{-1}$) | 0.0636 |
| | | $C_1$ | 7.1735 |
| | | $R_3^2$ | 0.9534 |

To predict the actual rate-controlling step in Levo-HCl adsorption and further explore the adsorption mechanism, a third model was applied. The intra-particle diffusion model provides a more comprehensive approach to defining the adsorption mechanism, including transport in the adsorbate, external mass transfer, diffusion in the pores and chemical reactions (adsorption–desorption). The model equation is expressed in equation (3.4):

$$q_t = k_3 t^{1/2} + C, \tag{3.4}$$

where $q_t$ is the adsorption amount at time $t$ (mg g$^{-1}$); $k_3$ is the rate constant for diffusion in the particle (mg (g min 1/2)$^{-1}$; its relationship with the internal diffusion coefficient $D$ is $k_3 = 6q_e/r\sqrt{D/\Pi}$, where $r$ is the particle radius); and $C$ is the thickness of the boundary layer, (mm).

The calculated kinetics parameters are shown in figure 13b and table 6. From the figure, the adsorption process of Levo-HCl by the CNCs-GO involves a multi-level linear relationship. Initially, the adsorption capacity increases rapidly with time. Then, the adsorption rate gradually decreases to a dynamic equilibrium, indicating that the adsorption process is affected by multiple diffusion steps. Three stages can be used to describe the diffusion/adsorption process for this system. The first stage is rapid adsorption, in which the Levo-HCl in the system quickly accumulates on the surface of the composite through membrane diffusion with a high adsorption rate. In the second stage, surface chemical adsorption occurs and the antibiotic molecules gradually enter into the adsorbent, which is a rapid diffusion process in the particles. As the reaction progresses, diffusion into the particles gradually slows due to the decreasing adsorbate concentration, an increasing mass transfer resistance and an increasing boundary layer effect. In the third stage, the Levo-HCl continues to interact with the adsorbent, diffusion in the particles decreases, the solid–liquid phase distribution gradually becomes balanced, and the adsorption amount no longer increases but gradually reaches equilibrium. The intra-particle diffusion model posits that, if three straight lines in the graphed equation pass through the coordinate origin, the rate-controlling step is intra-particle diffusion; if this condition does not hold, intra-particle diffusion is not the only controlling step, and other processes control the reaction rate, these processes together form control steps [56,57]. The three phase-fitting equations for the adsorption of Levo-HCl on the CNCs-GO do not pass through the origin, indicating that intra-particle diffusion is not the only rate-limiting step for the process, which therefore may be the combined effect of internal diffusion and surface adsorption.

## 3.4. Adsorption isotherms

The equilibrium adsorption isotherm is also used to understand the mechanism of the adsorption. Adsorption isotherm models describe the interaction between the adsorbate and adsorbent. Thus, nonlinear regression of the equilibrium data by either theoretical or empirical isotherm equations is essential to optimize the design of an adsorption system. In this paper, the isotherm parameters for the adsorption of Levo-HCl on CNCs-GO were obtained using three equations from classical isotherm models (Langmuir, Freundlich and Sips). The adsorption of Levo-HCl on CNCs-GO was analysed at different temperatures. The Langmuir isotherm model is based on single-layer adsorption on a surface with a limited number of adsorption points and uniform adsorption energy [58]. The equation is expressed as follows in equation (3.5):

$$q_e = \frac{K_L q_{max} C_e}{1 + K_L C_e},\tag{3.5}$$

where $q_e$ is the equilibrium adsorption capacity (mg g$^{-1}$); $C_e$ is the equilibrium concentration of the adsorbate in the solution (mg l$^{-1}$); $q_{max}$ is the maximum amount of Levo-HCl adsorbed per unit weight of the adsorbent (mg g$^{-1}$); and $K_L$ is the Langmuir adsorption equilibrium constant related to the affinity of the binding sites and indicates the bond energy of the adsorption reaction between adsorbent and adsorbate (l g$^{-1}$).

The Freundlich isotherm model is an empirical equation that is used to understand adsorption on heterogeneous surfaces with multiple adsorption layers [58]. The equation is expressed as follows in equation (3.6):

$$q_e = K_F C_e^{1/n},\tag{3.6}$$

where $K_F$ (mg g$^{-1}$) and $n$ are Freundlich constants related to the adsorption capacity and adsorption intensity and spontaneity, respectively. A value of $n$ in the range of $1 < n < 10$ indicates a favourable adsorption process. The greater the value of $n$, the more favourable is the adsorption.

The Sips model is a combination of the Langmuir and Freundlich isotherms [59] and is expressed in equation (3.7) as

$$q_e = q_{max} \times \frac{(K_S C_e)^\gamma}{1 + (K_S C_e)^\gamma},\tag{3.7}$$

where $q_{max}$ is the specific adsorption capacity at saturation (mg g$^{-1}$); $K_S$ is the Sips isotherm constant related to the energy of adsorption (ml mg$^{-1}$) and $\gamma$ is a heterogeneity factor. If the value of $K_S$ approaches zero, the Sips isotherm equation follows the Freundlich isotherm model, whereas if the value of $\gamma$ is equal or close to 1, the Sips isotherm equation reduces to the Langmuir isotherm.

Plots of the adsorption isotherms are illustrated in figure 14 and the calculated parameters are listed in table 7. The adsorption amount is found to increase rapidly when the equilibrium concentration is low. As the equilibrium concentration increases, the adsorption rate gradually decreases, and the adsorption amount gradually approaches a saturated state. This may be because, for a fixed CNCs-GO dosage, when the antibiotic concentration increases, more antibiotic molecules gather around the active sites of the CNCs-GO and the adsorption capacity increases. When the concentration exceeds a certain level, the surface-active sites become fully occupied and the CNCs-GO can no longer absorb additional antibiotic molecules; thus, the adsorption capacity is close to saturation and the removal rate decreases accordingly. Compared with Langmuir and Freundlich models, the correlation coefficient ($R^2$) of the Sips model fitting is closer to 1, indicating that the Sips isothermal model can describe the adsorption process well. In addition, in the Sips model, the value of $\gamma$ is often used to indicate the heterogeneity of adsorption. The closer the value of $\gamma$ is to 0, the more uneven the surface adsorption of the adsorbent is. In this study, the $\gamma$ of the Sips model is 0.3852, which is close to zero, indicating that the surface of CNCs-GO is not uniform, that is, the adsorption process is a multi-layer chemical adsorption on a heterogeneous surface.

## 3.5. Analysis of the interaction mechanism

For an additional investigation of the possible potential adsorption sites and bonding modes, XPS and SEM-EDS were used to detect the synthesized CNCs-GO after Levo-HCl adsorption. Figure 15a illustrates the XPS wide-scan spectra of the CNCs-GO composite after Levo-HCl adsorption. A distinct peak appeared after adsorption at a binding energy of approximately 685.1 eV and was

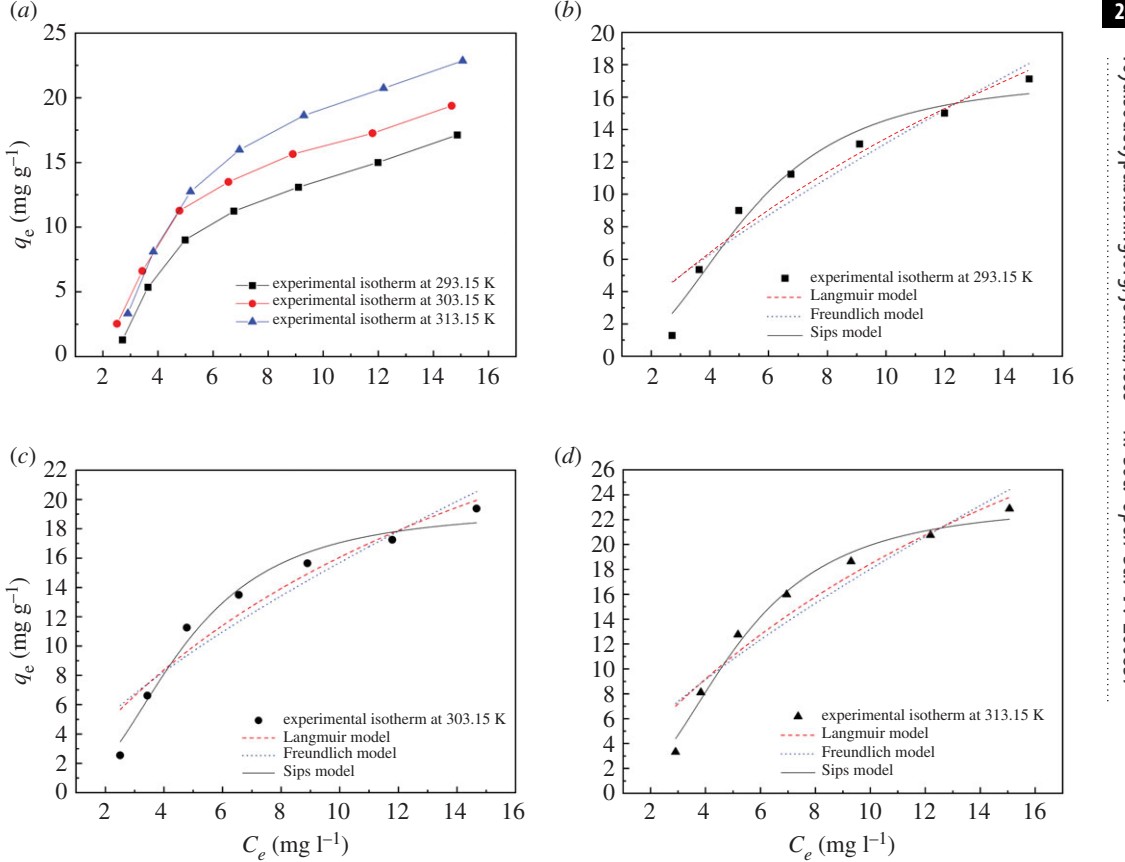

**Figure 14.** Adsorption isotherms of CNCs-GO for levofloxacin hydrochloride at different temperatures and fitting of Langmuir, Freundlich and Sips models.

**Table 7.** Isotherm parameters for adsorption of levofloxacin hydrochloride onto CNCs-GO at various temperatures.

| | temperature | | |
|---|---|---|---|
| isotherm model | 293.15 K | 303.15 K | 313.15 K |
| **Langmuir** | | | |
| $q_m$ (mg g$^{-1}$) | 49.7225 | 41.4427 | 55.7460 |
| $K_L$ (l mg$^{-1}$) | 0.0370 | 0.0633 | 0.0495 |
| $R^2$ | 0.9203 | 0.9262 | 0.9291 |
| **Freundlich** | | | |
| $K_F$ (mg$^{1-1/n}$ l$^{1/n}$ g$^{-1}$) | 2.0657 | 3.1220 | 3.2790 |
| $1/n$ | 0.8040 | 0.7016 | 0.7403 |
| $R^2$ | 0.9030 | 0.8980 | 0.9049 |
| **Sips** | | | |
| $q_m$ (mg g$^{-1}$) | 17.2922 | 19.3374 | 23.2898 |
| $K_s$ (l mg$^{-1}$) | 0.1911 | 0.2200 | 0.1975 |
| $\gamma$ | 0.3852 | 0.3918 | 0.3809 |
| $R^2$ | 0.9739 | 0.9825 | 0.9882 |

assigned to Levo-HCl, suggesting that the Levo-HCl was removed entirely. In addition, the composition of the composites after Levo-HCl adsorption was verified by SEM-EDS (figure 15b,c). The ratio of Leco-HCl was estimated at about 1.90% and is evenly dispersed on the surface of the material.

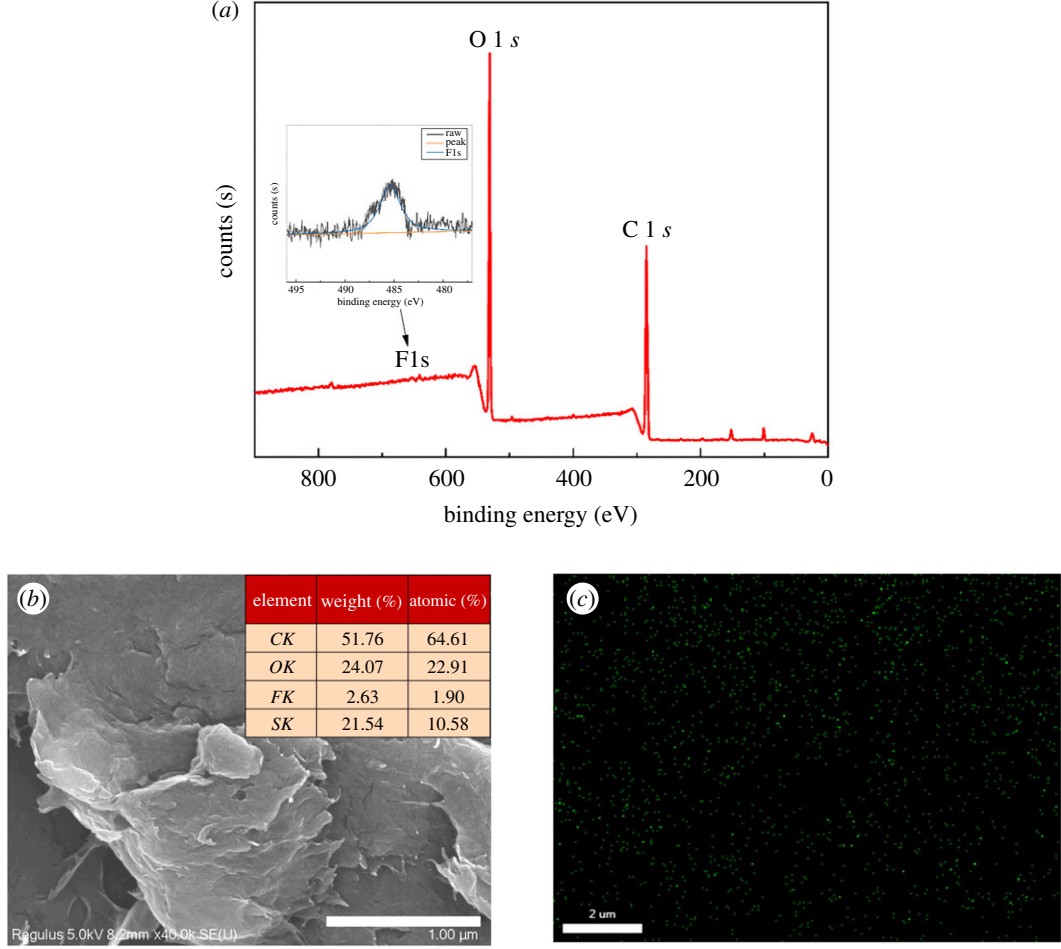

**Figure 15.** (*a*) XPS and (*b,c*) SEM-EDS spectra of CNCs-GO after Levo-HCl adsorption.

The possible adsorption mechanism of the antibiotics by CNCs-GO is based on the following properties. First, the composite material has a large surface area and many adsorption sites due to its large pore area and large void space between the core and the shell, which is conducive to the free diffusion of antibiotics on the adsorbent, and fully exposed active sites would enhance the opportunity for antibiotics to contact with the CNCs-GO via electrostatic attraction. Second, a lot of oxygen-containing functional groups both in CNCs-GO composite material afford the formation of hydrogen bonds with antibiotics molecules and rely on hydrogen bonding to produce adsorption. In addition, CNCs-GO with primarily π–π stacking could act as electron acceptors and be advantageous for adsorbing the antibiotics with an unsaturated bond (figure 16) [60]. Therefore, CNCs-GO can rapidly and efficiently adsorb levofloxacin hydrochloride antibiotics from aqueous solution due to their large specific surface area and abundant active, through sites π-π bond stacking, hydrogen bonding and electrostatic attraction.

# 4. Conclusion

In conclusion, a CNCs-GO was prepared ultrasonically and its adsorption properties for the antibiotic levofloxacin hydrochloride were studied. Based on single-factor tests, a multiple regression model was obtained through fitting with Design-Expert 10.0 software, and the reliability of the model was verified. According to the optimization results from response surface graphs and practical operation, the optimal conditions for adsorption were determined as an initial pollutant concentration of $10.0 \, \text{mg} \, \text{l}^{-1}$, an initial pH of 4, an adsorbent dosage of $0.1 \, \text{g} \, \text{l}^{-1}$ and a 4 h contact time. Adsorption kinetics data were best fitted with a pseudo-second-order model, suggesting the interaction of Levo-HCl with the CNCs-GO via hydrogen bonding as well as electronic interaction, and the adsorption is

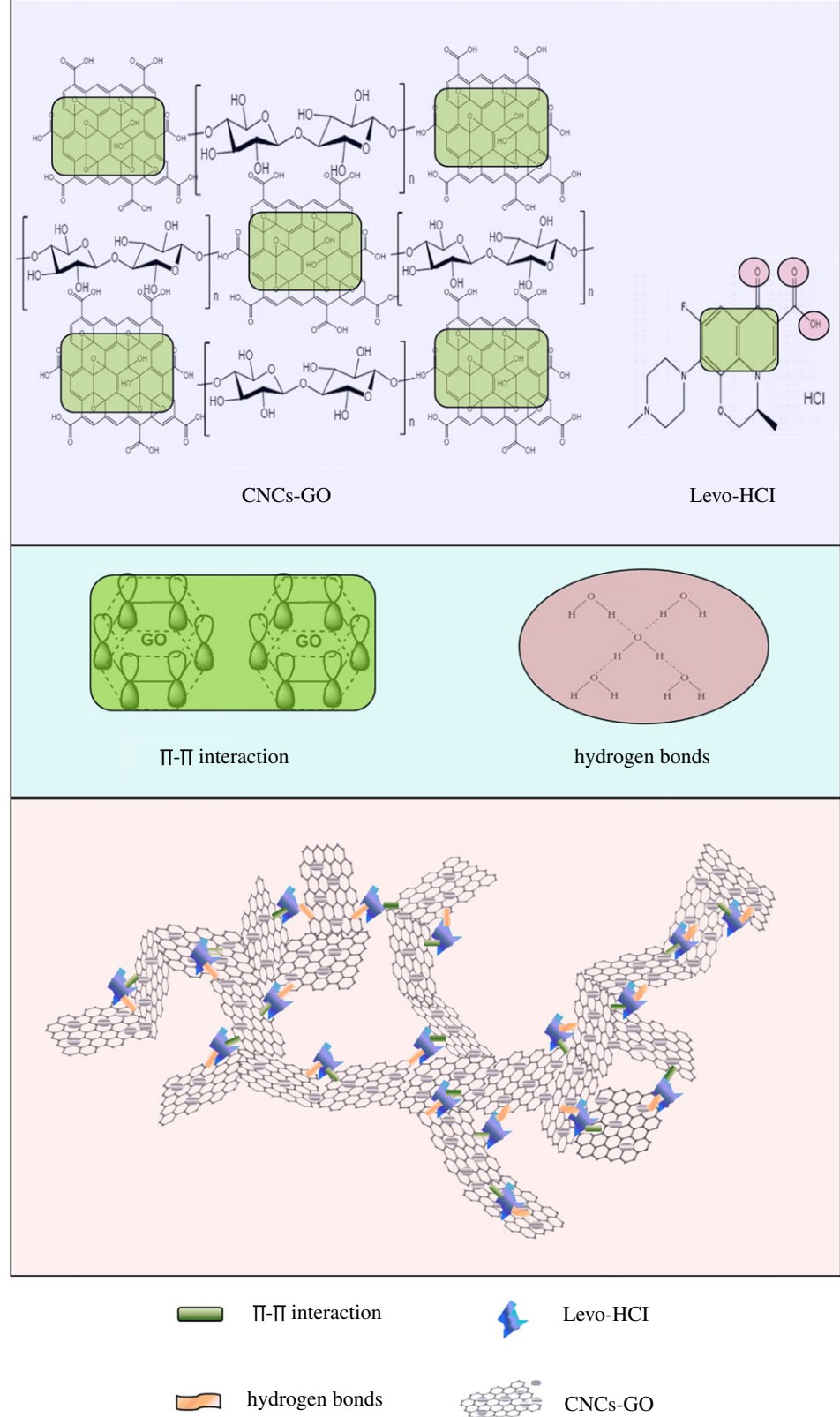

**Figure 16.** Schematic diagram of the CNCs-GO for antibiotics adsorption.

mainly controlled by chemical adsorption behaviour. The three-parameter Sips model well described the adsorption isotherms of the CNCs-GO. Finally, this research shows that the prepared CNCs-GO composite can be a potentially effective absorbent for the removal of antibiotics such as levofloxacin hydrochloride from aqueous solution.

**Ethics.** This study does not present research with ethical considerations.

**Data accessibility.** Our data are deposited at the Dryad Digital Repository: https://doi.org/10.5061/dryad.47d7wm39s [61].

**Authors' contributions.** J.T., J.Y. and C.M. were involved in conceptualization, methodology and software; J.T. and J.Y. were involved in data curation and writing the original draft preparation; K.D., Z.W. and C.C. were involved in visualization and investigation; J.L. was involved in supervision; J.L. and Z.W. were involved in software and validation; J.L., Z.W. and C.Z. were involved in reviewing and editing.

**Competing interests.** The authors declare no competing of interests.

**Acknowledgements.** Financial support from the National Natural Science Foundation of China (grant no. U1803244), the National Key R&D Program of China (grant no. 2017YFC0404304), the Key Science and Technology Project in special issues of Bingtuan (grant nos. 2019DB007 and 2019AB035) and the National Natural Science Foundation of China (grant no. 51609260) are gratefully acknowledged.

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
