## [Reviewer comments · Royal Society Open Science]

Review History

RSOS-200857.R0 (Original submission)

Review form: Reviewer 1

Is the manuscript scientifically sound in its present form?

Yes

Are the interpretations and conclusions justified by the results?

Yes

Is the language acceptable?

Yes

Do you have any ethical concerns with this paper?

No

Have you any concerns about statistical analyses in this paper?

Yes

Recommendation?

Accept with minor revision (please list in comments)

Comments to the Author(s)

Manuscript Number: RSOS-200857

Journal: Royal Society Open Science

Title :

Cellulose nanocrystals/graphene oxide composite for the adsorption and removal of levofloxacin hydrochloride antibiotic from aqueous solution

Comments and Recommendation:

It is very important to obtain good view study the physical structure and chemical 7 properties of the CNCs-GO. The three-dimensional structure of the composite material rendered a 8 high surface area and electrostatic attraction, resulting in an increased adsorption capacity of the 9 CNCs-GO for Levo-HCl. Based on the Box-Behnken design, the effects of different factors on the 10 removal of Levo-HCl by the CNCs-GO were explored.I have some comments as following:

1. In introduction , Authors must be reduce the general parts and focus on aim of the work
2. Author should add more reference in introduction part in adsorption part and Central composite statistical analysis part as Melamine grafted chitosan-montmorillonite nanocomposite for ferric ions adsorption: Central composite design optimization study, Journal of Cleaner Production,Volume 241, 20 December 2019, 118189
3. Authors should make more characterization as GPC, TEM , DLS , Elemental analysis , mass spectroscopy for composites .
4. Authors should rescan of SEM because magnification powers to are different.

Recommendation: minor revision

Best regards

Review form: Reviewer 2

Is the manuscript scientifically sound in its present form?

Yes

Are the interpretations and conclusions justified by the results?

Yes

Is the language acceptable?

Yes

Do you have any ethical concerns with this paper?

No

Have you any concerns about statistical analyses in this paper?

No

Recommendation?

Major revision is needed (please make suggestions in comments)

Comments to the Author(s)

The manuscript entitled " Cellulose nanocrystals/graphene oxide composite for the adsorption and removal of levofloxacin hydrochloride antibiotic from aqueous solution " written by Tao,

Junhong et.al, reported that cellulose nanocrystals/graphene oxide composite is used to remove levofloxacin hydrochloride antibiotic from aqueous solution. The materials reported in the manuscript were analyzed by SEM, XPS and FT-IR. In my view, there are some questions need to be solved before it can be accepted for publication. Here is the detail of necessary revision,

1. Please add the reagent grade of "levofloxacin hydrochloride, cellulose nanocrystals, and graphite powder" in the 2.1 materials section.
2. Compared with HPLC, UV detection method, it is more convenient and fast ,and can also achieve a certain degree of accuracy. Please explain why UV is not selected to detect the concentration of levofloxacin.
3. The coordinates of the pictures in figure 8 of the manuscript are not clear and cannot be seen clearly.
4. Page 21 line 44-48: "The influence of the adsorbent concentration and initial solution pH on the removal rate under the center point conditions of the pollutant concentration (10.0 mg·L⁻¹) and contact time (4 h) is shown in Error! Reference source not found. a." There are many similar mistakes in this article, please check and correct them.
5. Page 30 line 53-59: "The equilibrium experimental data was fitted using the Sips adsorption isotherm model. The correlation coefficient of the fitted results was R²> 0.98, and the non-uniformity coefficient $\gamma = 0.3852$, indicating that the adsorption type of antibiotics on CNCs-GO was non-homogeneous monolayer chemisorption." Please explain this sentence.
6. It is suggested that the author should supplement the comparative diagram of SEM and XPS before and after adsorption.

Decision letter (RSOS-200857.R0)

Dear Dr LI:

Title: Cellulose nanocrystals/graphene oxide composite for the adsorption and removal of levofloxacin hydrochloride antibiotic from aqueous solution
 Manuscript ID: RSOS-200857

The editor assigned to your manuscript has now received comments from reviewers. We would like you to revise your paper in accordance with the referee and Subject Editor suggestions which can be found below (not including confidential reports to the Editor). Please note this decision does not guarantee eventual acceptance.

Please submit your revised paper before 15-Jul-2020. Please note that the revision deadline will expire at 00.00am on this date. If we do not hear from you within this time then it will be assumed that the paper has been withdrawn. In exceptional circumstances, extensions may be possible if agreed with the Editorial Office in advance. We do not allow multiple rounds of revision so we urge you to make every effort to fully address all of the comments at this stage. If deemed necessary by the Editors, your manuscript will be sent back to one or more of the original reviewers for assessment. If the original reviewers are not available we may invite new reviewers.

To revise your manuscript, log into <http://mc.manuscriptcentral.com/rsos> and enter your Author Centre, where you will find your manuscript title listed under "Manuscripts with Decisions." Under "Actions," click on "Create a Revision." Your manuscript number has been

appended to denote a revision. Revise your manuscript and upload a new version through your Author Centre.

On behalf of the Subject Editor Professor Anthony Stace and the Associate Editor Dr Darren Walsh.

RSC Associate Editor:
Comments to the Author:
(There are no comments.)

RSC Subject Editor:
Comments to the Author:
(There are no comments.)

Reviewers' Comments to Author:
Reviewer: 1

Comments to the Author(s)
Manuscript Number: RSOS-200857

Journal: Royal Society Open Science

Title :
Cellulose nanocrystals/graphene oxide composite for the adsorption and removal of levofloxacin hydrochloride antibiotic from aqueous solution

Comments and Recommendation:
It is very important to obtain good view study the physical structure and chemical 7 properties of the CNCs-GO. The three-dimensional structure of the composite material rendered a 8 high surface area and electrostatic attraction, resulting in an increased adsorption capacity of the 9 CNCs-GO for Levo-HCl. Based on the Box-Behnken design, the effects of different factors on the 10 removal of Levo-HCl by the CNCs-GO were explored. I have some comments as following:

1. In introduction , Authors must be reduce the general parts and focus on aim of the work
2. Author should add more reference in introduction part in adsorption part and Central composite statistical analysis part as Melamine grafted chitosan-montmorillonite nanocomposite for ferric ions adsorption: Central composite design optimization study, Journal of Cleaner Production, Volume 241, 20 December 2019, 118189
3. Authors should make more characterization as GPC, TEM , DLS , Elemental analysis , mass spectroscopy for composites .
4. Authors should rescan of SEM because magnification powers to are different.

Recommendation: minor revision
Best regards

Reviewer: 2

Comments to the Author(s)

The manuscript entitled " Cellulose nanocrystals/ graphene oxide composite for the adsorption and removal of levofloxacin hydrochloride antibiotic from aqueous solution " written by Tao, Junhong et.al, reported that cellulose nanocrystals/ graphene oxide composite is used to remove levofloxacin hydrochloride antibiotic from aqueous solution. The materials reported in the manuscript were analyzed by SEM, XPS and FT-IR. In my view, there are some questions need to be solved before it can be accepted for publication. Here is the detail of necessary revision,

1. Please add the reagent grade of "levofloxacin hydrochloride, cellulose nanocrystals, and graphite powder" in the 2.1 materials section.
2. Compared with HPLC, UV detection method, it is more convenient and fast ,and can also achieve a certain degree of accuracy. Please explain why UV is not selected to detect the concentration of levofloxacin.
3. The coordinates of the pictures in figure 8 of the manuscript are not clear and cannot be seen clearly.
4. Page 21 line 44-48: "The influence of the adsorbent concentration and initial solution pH on the removal rate under the center point conditions of the pollutant concentration (10.0 mg·L⁻¹) and contact time (4 h) is shown in Error! Reference source not found. a." There are many similar mistakes in this article, please check and correct them.
5. Page 30 line 53-59: "The equilibrium experimental data was fitted using the Sips adsorption isotherm model. The correlation coefficient of the fitted results was R²> 0.98, and the non-uniformity coefficient $\gamma = 0.3852$, indicating that the adsorption type of antibiotics on CNCs-GO was non-homogeneous monolayer chemisorption." Please explain this sentence.
6. It is suggested that the author should supplement the comparative diagram of SEM and XPS before and after adsorption.

Author's Response to Decision Letter for (RSOS-200857.R0)

See Appendix A.

RSOS-200857.R1 (Revision)

Review form: Reviewer 1

Is the manuscript scientifically sound in its present form?

Yes

Are the interpretations and conclusions justified by the results?

Yes

Is the language acceptable?

Yes

Do you have any ethical concerns with this paper?

No

Have you any concerns about statistical analyses in this paper?

No

Recommendation?

Accept as is

Comments to the Author(s)

all comments done

Decision letter (RSOS-200857.R1)

Dear Dr LI:

Title: Cellulose nanocrystals/graphene oxide composite for the adsorption and removal of levofloxacin hydrochloride antibiotic from aqueous solution
Manuscript ID: RSOS-200857.R1

It is a pleasure to accept your manuscript in its current form for publication in Royal Society Open Science. The chemistry content of Royal Society Open Science is published in collaboration with the Royal Society of Chemistry. I apologise it has taken longer than usual to send you this decision.

Yours sincerely,

Dr Laura Smith

Publishing Editor, Journals

On behalf of the Subject Editor Professor Anthony Stace and the Associate Editor Dr Darren Walsh.

RSC Associate Editor:
Comments to the Author:
(There are no comments.)

RSC Subject Editor:
Comments to the Author:
(There are no comments.)

Reviewer(s)' Comments to Author:
Reviewer: 1

Comments to the Author(s)
all comments done

Appendix A

Dear Editor and Reviewers:

Thanks for your letter and the reviewer's comments concerning our manuscript "*Cellulose nanocrystals/graphene oxide composite for the adsorption and removal of levofloxacin hydrochloride antibiotic from aqueous solution*". These comments are all valuable and very helpful for revising and improving our paper, as well as the important guiding significance to us. We have studied the comments very carefully and tried our best to revise our manuscript according to the comments. The related changes in the revised manuscript are in *blue* color. The detailed modifications in the article and the responses to the Editorial Board Member's and the reviewer's comments are as following:

Reviewer #1

Comments: It is very important to obtain good view study the physical structure and chemical 7 properties of the CNCs-GO. The three-dimensional structure of the composite material rendered a 8 high surface area and electrostatic attraction, resulting in an increased adsorption capacity of the 9 CNCs-GO for Levo-HCl. Based on the Box–Behnken design, the effects of different factors on the 10 removal of Levo-HCl by the CNCs-GO were explored. I have some comments as following:

1. In introduction, Authors must be reduce the general parts and focus on aim of the work.

Author reply:

Thank you very much for the precious suggestion. Under the reviewer's suggestion, in the introduction, the general part is reduced, the sentence for the purpose of the work is added, and the last paragraph of the introduction is rewritten.

The following sentence has been added on **Line 48-49**:

"Hence, nanocellulose nanocomposites have been used as sorbents for heavy metals and organic pollutants removal from aqueous solution. [1]"

The following sentence has been added on **Line 58-60**:

"In recent, GO has attracted increasing attention as new adsorbent owing to its distinguished properties of high surface area as well as easy to functionalize ability. [2]"

We have rewritten the last paragraph of the introduction, and the last paragraph of the introduction is as follows:

"The purpose of the present work is the synthesis of novel adsorbent, cellulose nanocrystals/graphene oxide nanocomposite (CNCs-GO), for the removal of antibiotic

levofloxacin hydrochloride (Levo-HCl, Table 1), a pharmaceutical contaminant in waste water treatment. Adsorbent was characterized using FTIR, XRD, SEM-EDS, XPS and BET analysis. The response surface method (RSM) was used to optimize Levo-HCl adsorption conditions. Finally, we investigated the adsorption performance and mechanism for Levo-HCl removal process by CNCs-GO.”

Reference:

- [1] Mahfoudhi, N. and S. Boufi, Nanocellulose as a novel nanostructured adsorbent for environmental remediation: a review. *Cellulose*, 2017. 24(3): p. 1171-1197.
 - [2] Upadhyay, R.K., N. Soin, and S.S. Roy, Role of graphene/metal oxide composites as photocatalysts, adsorbents and disinfectants in water treatment: a review. *Rsc Advances*, 2014. 4(8): p. 3823-3851.
2. Author should add more reference in introduction part in adsorption part and Central composite statistical analysis part as Melamine grafted chitosan-montmorillonite nanocomposite for ferric ions adsorption: Central composite design optimization study, *Journal of Cleaner Production*, Volume 241, 20 December 2019, 118189.

Author reply:

Thanks for your valuable comments and kind suggestion. The above literature is very relevant to the contents of this manuscript. According to your suggestion, we have add the literature, and this document is cited in the introduction, adsorption, and response surface analysis. The document number is [32].

3. Authors should make more characterization as GPC, TEM, DLS, Elemental analysis, mass spectroscopy for composites.

Author reply:

We are grateful for the pertinent comment of the reviewer. According to the reviewer's comments, we supplemented the TEM and DLS characterization of CNCs-GO composites.

The following sentence has been added on **Line 148-152**:

“Fig. 2 demonstrates the TEM micrographs of CNCs-GO. From the TEM micrograph, it can be seen that nano-sized CNCs and GO's CNCs-GO composites are formed. The CNCs are evenly

distributed between the GO slices, interlaced with each other. At the same time, there are fewer folds of GO sheets, which may be due to the interaction between CNCs and GO, which prevents the folding of GO sheets.”

Fig. 2 TEM images of CNCs-GO

The following sentence has been added on **Line 173-182**:

“Dynamic light scattering (DLS) measures particle size on the basis of fluctuations in scattered light intensity with time that may be due to the random Brownian motion of the sample particles present in suspension or polymers in a solution. Diffusion is directly related to statistical nature of these fluctuations in scattered intensity.^[1, 2] In addition, dynamic light scattering can be used to help prove whether the prepared composite material is at the nano-scale level.^[3] As shown in Fig. 4, the CNCs-GO were unimodal in distribution, all the particles are in a range between 300 and 1000 d. nm, and the mean average size of the resulting nanoparticles was found to be 842.3 (d.nm), showing highly stable without forming any aggregation. At the same time, a narrow size distribution indicate many particles are homogeneous in size with small variations, this result proves that the composite particles have a nano-size.”

Fig. 4 Spectroscopic measurement of particle size by dynamic light scattering of CNCs-GO

Reference:

- [1] Hoo, C.M., et al., A comparison of atomic force microscopy (AFM) and dynamic light scattering (DLS) methods to characterize nanoparticle size distributions. *Journal of Nanoparticle Research*, 2008. 10: p. 89-96.
- [2] Yang, Y.J., et al., Particle agglomeration and properties of nanofluids. *Journal of Nanoparticle Research*, 2012. 14(5).
- [3] Mandal, A. and D. Chakrabarty, Isolation of nanocellulose from waste sugarcane bagasse (SCB) and its characterization. *Carbohydrate Polymers*, 2011. 86(3): p. 1291-1299.

4. Authors should rescan of SEM because magnification powers to are different.

Author reply:

According to the reviewer's comments, we re-selected the appropriate and clear SEM images. As shown below. The SEM image contains photos of CNCs, GO and CNCs-GO at 5000× and 30,000× magnifications respectively.

Fig. 1 SEM images of CNCs, GO and CNCs-GO: (a) SEM images of CNCs at 5000×magnifications; (b) SEM images of CNCs at 30000×magnifications; (c) SEM images of GO at 5000×magnifications;(d) SEM images of GO at 30000×magnifications; (e) SEM images of CNCs-GO at 5000×magnifications;(f) SEM images of CNCs-GO at 30000×magnifications

Reviewer #2

Comments: The manuscript entitled "Cellulose nanocrystals/graphene oxide composite for the adsorption and removal of levofloxacin hydrochloride antibiotic from aqueous solution" written by Tao, Junhong et.al, reported that cellulose nanocrystals/graphene oxide composite is used to remove levofloxacin hydrochloride antibiotic from aqueous solution. The materials reported in the manuscript were analyzed by SEM, XPS and FT-IR. In my view, there are some questions need to be solved before it can be accepted for publication. Here is the detail of necessary revision,

1. Please add the reagent grade of “levofloxacin hydrochloride, cellulose nanocrystals, and graphite powder” in the 2.1 materials section.

Author reply:

Thanks for your valuable suggestion. We have rewritten the 2.1 materials section based on your suggestion, and the 2.1 materials section is as follows.

2.1 Materials

Levofloxacin hydrochloride 98%, cellulose nanocrystals, and graphite powder 99.95% (for the preparation of graphene oxide) were purchased from Shanghai Macklin Biochemical Co., Ltd. Deionized water was used for all experiments. Otherwise specified chemicals were of reagent grade and utilized without further purification. Deionized water was used throughout the experiments.

2. Compared with HPLC, UV detection method, it is more convenient and fast, and can also achieve a certain degree of accuracy. Please explain why UV is not selected to detect the concentration of levofloxacin.

Author reply:

We really appreciate the reviewer's precious comments, which is very important for us to improve our work. High performance liquid chromatography (HPLC) and ultraviolet-visible spectrophotometry (UV-Vis) are two commonly used methods for detecting Levofloxacin. Wang investigated the high-performance liquid chromatography and ultraviolet-visible spectrophotometry to determine the concentration of levofloxacin hydrochloride on a new synthetic scaffold, and compared the characteristics of the two detection methods. The findings demonstrated that although UV-Vis is simple to operate, and the expenses are low, it is not accurate to measure the concentration of drugs loaded on the composite composites. The HPLC method could be used to separate Levofloxacin from various impurities in the chromatographic column, which eliminated the interference of impurities with Levofloxacin. As a method, HPLC exhibited the advantages of high precision and high recovery.^[1]

In addition, some of the literature on the removal of antibiotics also uses high-performance liquid chromatography to determine the concentration of antibiotics,^[2-6] and our laboratory has the conditions to use high performance liquid chromatography, so in this study, high performance liquid chromatography was used to determine antibiotics.

Reference

- [1] Wang, Q., et al., Comparison of high-performance liquid chromatography and ultraviolet-visible spectrophotometry to determine the best method to assess Levofloxacin released from mesoporous silica microspheres/nano-hydroxyapatite composite scaffolds. *Experimental and Therapeutic Medicine*, 2019. **17**(4): p. 2694-2702.

- [2] Song, Y.X., et al., Nanocomposites of zero-valent Iron@Activated carbon derived from corn stalk for adsorptive removal of tetracycline antibiotics. *Chemosphere*, 2020. 255.
- [3] Chen, S., et al., Ecotoxicological effects of sulfonamides and fluoroquinolones and their removal by a green alga (*Chlorella vulgaris*) and a cyanobacterium (*Chrysochloris ovalisporum*). *Environmental Pollution*, 2020. 263.
- [4] Malakootian, M., A. Nasiri, and M.A. Gharaghani, Photocatalytic degradation of ciprofloxacin antibiotic by TiO₂ nanoparticles immobilized on a glass plate. *Chemical Engineering Communications*, 2020. 207(1): p. 56-72.
- [5] Li, N., et al., Simultaneous removal of tetracycline and oxytetracycline antibiotics from wastewater using a ZIF-8 metal organic-framework. *Journal of Hazardous Materials*, 2019. 366: p. 563-572.
- [6] Ahmed, M.B., et al., Single and competitive sorption properties and mechanism of functionalized biochar for removing sulfonamide antibiotics from water. *Chemical Engineering Journal*, 2017. 311: p. 348-358.

3. The coordinates of the pictures in figure 8 of the manuscript are not clear and cannot be seen clearly.

Author reply:

According to the reviewer's comments, the Fig. 8 and Fig. 9 have been changed and kept in better image quality.

4. Page 21 line 44-48: "The influence of the adsorbent concentration and initial solution pH on the removal rate under the center point conditions of the pollutant concentration (10.0 mg·L⁻¹) and contact time (4 h) is shown in Error! Reference source not found." There are many similar mistakes in this article, please check and correct them.

Author reply:

Many thanks for the valuable suggestion and criticism. We have solved the problem of show in "Error! Reference source not found".

5. Page 30 line 53-59: "The equilibrium experimental data was fitted using the Sips adsorption

isotherm model. The correlation coefficient of the fitted results was $R^2 > 0.98$, and the non-uniformity coefficient $\gamma = 0.3852$, indicating that the adsorption type of antibiotics on CNCs-GO was non-homogeneous monolayer chemisorption.” Please explain this sentence.

Author reply:

We are grateful for the pertinent comment of the reviewer. Under the reviewer’s suggestion, we found the wrong conclusion in this paragraph. According to the reviewer’s suggestion, we have explained the sentence. Sips model is a hybrid model based on Langmuir and Freundlich models. Sips model is often used to represent the adsorption of non-uniform surfaces. In the Sips equation, γ is a heterogeneous coefficient, which is often used to indicate the heterogeneity of adsorption. If the γ value approaches zero, indicating that the surface adsorption is more uneven, the Sips isotherm equation follows the Freundlich isotherm model; when the γ value is equal to or close to 1, indicating that the surface adsorption is more uniform, the Sips isotherm equation is simplified to Langmuir isotherm model^[1-2]. Among the three isothermal models, the Sips isothermal model has the largest correlation coefficient ($R^2 > 0.98$), indicating that the Sips isothermal model can describe the adsorption process well. The heterogeneity coefficient $\gamma = 0.3852$, approaching zero, that is, the adsorption process in the low concentration antibiotic solution is the multilayer adsorption of the heterogeneous surface of the Freundlich model.

The erroneous conclusion we found is " the non-uniformity coefficient $\gamma = 0.3852$, indicating that the adsorption type of antibiotics on CNCs-GO was non-homogeneous monolayer chemisorption". The correct conclusion should be "multimolecular layer chemisorption on heterogeneous surface".

We have partially rewritten the paragraph, and the paragraph is as follows :

“Compared with Langmuir and Freundlich models, the correlation coefficient (R^2) of the Sips model fitting is closer to 1, indicating that the Sips isothermal model can describe the adsorption process well. In addition, in the Sips model, the value of γ is often used to indicate the heterogeneity of adsorption. The closer the value of γ is to 0, the more uneven the surface adsorption of the adsorbent is. In this study, the γ of the Sips model is 0.3852, which is close to zero, indicating that the surface of CNCs-GO is not uniform, that is, the adsorption process is a multi-layer chemical adsorption on a heterogeneous surface.”

Reference

- [1] Alatalo Sara-Maaria, Repo Eveliina, Mäkilä Ermei, et al. Adsorption behavior of hydrothermally treated municipal sludge & pulp and paper industry sludge. *Bioresource Technology*. 2013, 147:71-76
- [2] Jeppu Gautham P., Clement T. Prabhakar. A modified Langmuir-Freundlich isotherm model for simulating pH-dependent adsorption effects. *Journal of Contaminant Hydrology*. 2012, 129-130(3):46-53.

6. It is suggested that the author should supplement the comparative diagram of SEM and XPS before and after adsorption.

Author reply:

Thanks for your valuable comments and kind suggestion. According to the reviewer's comments, we supplemented the SEM-EDS and XPS characterization of the CNCs-GO composite after adsorption of Levo-HCl.

The following sentence has been added on **Line 583-590**:

“For an additional investigation of the possible potential adsorption sites and bonding modes, XPS and SEM-EDS were used to detect the synthesized CNCs-GO after Levo-HCl adsorption. Fig. 15a illustrates the XPS wide-scan spectra of the CNCs-GO composite after Levo-HCl adsorption. A distinct peak appeared after adsorption at a binding energy of approximately 685.1 eV and was assigned to Levo-HCl, suggesting that the Levo-HCl was removed entirely. In addition, the composition of the composites after Levo-HCl adsorption was verified by SEM-EDS (Fig. 15b, c). The ratio of Levo-HCl was estimated at about 1.90% and is evenly dispersed on the surface of the material.”

Fig. 15 (a) XPS and (b) (c) SEM-EDS spectra of CNCs-GO after Levo-HCl adsorption.